# Protocols for Verifying Smooth Strategies in Bandits and Games

**Miranda Christ**
Columbia University
mchrist@cs.columbia.edu

**Daniel Reichman**
Worcester Polytechnic Institute
daniel.reichman@gmail.com

**Jonathan Shafer**
MIT
shaferjo@mit.edu

## Abstract

We study protocols for verifying approximate optimality of strategies in multi-armed bandits and normal-form games. As the number of actions available to each player is often large, we seek protocols where the number of queries to the utility oracle is *sublinear* in the number of actions. We prove that such verification is possible for sufficiently *smooth* strategies that do not put too much probability mass on any specific action and provide protocols for verifying that a smooth policy for a multi-armed bandit is close to optimal. Our verification protocols require provably fewer arm queries than learning. Furthermore, we show how to use cryptographic tools to reduce the communication cost of our protocols. We complement our protocol by proving a nearly tight lower bound on the query complexity of verification in our settings. As an application, we use our bandit verification protocol to build a protocol for verifying approximate optimality of a strong smooth Nash equilibrium, with sublinear query complexity.

## 1 Introduction

An overabundance of available actions makes decision-making challenging in numerous settings, from consumer choice to logistical operations and public policy. To gain information about the (often stochastic) reward an action entails, an agent can simply perform that action and experience the outcome. However, in many situations it is infeasible for an agent with limited resources to try even a small fraction of the set of available actions. One method to mitigate such choice overload is to rely on an external source providing information about the utility of actions. For example, one may consult recommendations from other customers when deciding whether to purchase a product, or solicit advice from consulting firms when formulating a business strategy or public policy. This information however, may be unreliable, and it is not clear how to efficiently *verify* its accuracy. Namely, can one outsource information collection to an untrusted external party, and verify correctness by independently assessing the utility of just a few actions?

Verifying utility estimates arises in machine learning (ML) applications as well. For example, an end user may delegate ML training to a better-resourced but untrusted external party [Goldwasser et al., 2021, 2015]. In *reinforcement learning*, a training algorithm might return a set of estimates for the rewards of possible actions. Once again, a challenge for the user is to verify the fidelity of these estimates, which could be inaccurate for numerous reasons. For example, the external party could be malicious, careless and susceptible to errors, or the training algorithm may be faulty. To address this challenge, a line of research develops learning methods that are robust to adversarial data corruption [Charikar et al., 2017, Canetti and Karchmer, 2021]. The specific case of corrupted rewards (utilities) in multi-armed bandits has received attention as well [Jun et al., 2018, Zhang et al., 2022].

Similar verification issues emerge in multiagent settings studied in game theory, where each agent's utility may depend on the other agents' actions. An object of central interest in such settings is the

39th Conference on Neural Information Processing Systems (NeurIPS 2025).

($\varepsilon$-approximate) *Nash equilibrium* (NE or $\varepsilon$-NE), which is a strategy profile[1] such that no agent can improve their own utility (by more than $\varepsilon$) by deviating unilaterally to another strategy. There is a host of questions related to verification and Nash equilibria; specifically, the current paper is motivated by scenarios such as:

- *An agent might want to verify that, given that the current strategies of the other agents are known, a proposed strategy is (approximately) the best possible.* For instance, suppose Alice is a corporation navigating a complex market, and she receives advice from a consulting firm that recommends a specific strategy $\tilde{\pi}$. Alice might subsequently like to verify that the proposed $\tilde{\pi}$ is indeed an (approximately) optimal strategy given current market conditions (i.e., given what other market participants are doing).

- *An agent might want to verify that, together with the known current strategy profile of the other agents, a proposed strategy forms an (approximate) Nash equilibrium.* For instance, in the previous example, assume the proposed strategy $\tilde{\pi}$ indeed has optimal expected utility for Alice in current market conditions. Nonetheless, adopting strategy $\tilde{\pi}$ might not be a good idea if, in the case where Alice follows strategy $\tilde{\pi}$, some other market participant, Bob, can unilaterally deviate to a strategy that benefits Bob but significantly damages Alice. Therefore, Alice might like to verify that, together with the current known strategy profile of the other market participants, the proposed strategy $\tilde{\pi}$ forms an (approximate) Nash equilibrium. That would be consistent with $\tilde{\pi}$ being a reasonable strategy for Alice to commit to.

We consider two popular models for problems of choosing between different actions: *multi-armed bandits* (MAB) Lattimore and Szepesvári [2020] in the single-agent setting, and *normal-form games* in the multi-agent setting. For both MAB and games, we consider the settings where $n$, the number of actions available to each player, is a large integer. The large number of actions, along with the prohibitive computational cost of computing approximate NE [Daskalakis et al., 2009, Rubinstein, 2017], motivates the reliance on advice from external parties. This leads to the following questions:

**Question 1.** *Given an MAB with $n$ arms, can we* verify *whether a list of purported rewards is close to the true rewards and output a near-optimal strategy, using a number of bandit queries that is sublinear in $n$?*

**Question 2.** *Given a $k$-player, $n$-action normal-form game, can a prover convince a verifier that a strategy profile is an $\varepsilon$-NE, with the verifier making a number of payoff oracle queries that is sublinear in $n$?*

We address these questions within the framework of *interactive proof systems* [Goldwasser et al., 1989]. In this setting, a computationally bounded verifier interacts with an untrusted prover, and can verify the correctness of statements that are intractable to assess without interaction [Shamir, 1992]. A recent development in the study of interactive proofs is the application of this methodology to machine learning [Goldwasser et al., 2021]. We adopt this development in our setting of verification in MABs. MABs are a fundamental concept in machine learning and are used in numerous theoretical and applied settings. Therefore, we hope that our study could lead to further discoveries regarding the potential uses of interactive proofs in machine learning and operations research. We focus on the *offline* setting of MABs and do not consider regret minimization. Studying interactive proofs for the online setting is an interesting direction for future work.

Let us first give a simple example where the verification of an MAB can be significantly more efficient compared to finding an optimal or near optimal strategy. Consider the promise problem where an MAB is known to have a single arm whose reward a deterministic reward of $1$, and all other $n-1$ arms have deterministic reward of $0$. It is trivial to *verify* that a given arm communicated by a prover guarantees a reward of $1$ using a single pull to that arm. On the other hand, *finding* such an arm can require $\Omega(n)$ pulls to the arms: if the single arm with reward one is chosen uniformly out of $n$ arms, a linear number of queries to the arms of the bandits is needed to find it [Kearns et al., 2002, Mannor and Tsitsiklis, 2004b].

Similarly, for a given parameter $\gamma$ it can be verified with probability at least $2/3$ that an MAB (with rewards in $[-1, 1]$) has an arm with expected reward at least $\gamma - \varepsilon$ for some accuracy parameter $\varepsilon$, using $O(1/\varepsilon^2)$ queries to an arm provided by a prover. As before, finding a strategy with an expected reward of $\gamma$ requires $\Omega(n)$ queries to the arms of the MAB.

---

[1]A *strategy* is a distribution over actions. A *strategy profile* is an assignment of a strategy to each player (i.e., agent) in the game.

Observe that in the first example, we have a promise problem in the sense that we know that there exists a single arm with reward 1 and that this is the arm with the highest reward. In the second example, we verify a *lower bound* on the value of an optimal strategy, but we do not verify an upper bound on the best possible value, [2] hence we do not verify that a given strategy is nearly optimal. This is not a coincidence: Even in the interactive proof setting, we show that $\Omega(n/\varepsilon^2)$ queries are necessary. Therefore to achieve verification with a sublinear number of queries to the arms of the bandits, we need to make additional assumptions on either the MAB, or the set of strategies considered. Here we show that verification with a sublinear number of queries to the arms of the bandit is possible for *smooth strategies* discussed in the next section.

In interactive proofs, the number of bits communicated between the prover and the verifier may become a bottleneck. Therefore, it is desirable to have *communication efficient* protocols where the number of bits communicated is sublinear in $n$ as well. We consider communication-efficient protocols in the setting where a prover wishes to convince the verifier of the value of the optimal policy, but does not need to send the optimal policy itself. We show that such communication-efficient protocols can be achieved for sufficiently smooth MABs by using cryptographic primitives such as vector commitments and SNARKs. While our basic protocol consists of just a single message sent from the prover to the verifier, our protocol with a sublinear number of bits of communication uses multiple rounds of interaction.

## 2 Verifying smooth strategies: model and results

### 2.1 Smooth strategies: motivation and background

A smooth strategy is a distribution over actions that is not too concentrated on any one action. More precisely, a strategy is $\sigma$-smooth if each action is selected with probability at most $\sigma$. Clearly $\sigma \in [1/n, 1]$, with smaller values of $\sigma$ corresponding to smoother (i.e., closer to uniform) distributions. To ensure that our protocols have sublinear query complexity, it is needed that $\sigma n = o(n)$. This is a natural assumption to ensure that the distribution is "well spread" in the sense that the support size of the distribution tends to infinity with $n$.

As a motivation, in many applications each arm in an MAB represents some resource that a system can utilize (e.g., a server in a data center, a driver in a ride-hailing service). Often, as the system grows, the number of available arms increases, as does the total number of arm-pulls, but the capacity of each individual arm remains bounded (e.g., as a ride-hailing service grows, the total number of rides and drivers increase, but the number of rides each individual driver can handle remains the same). If the total number of arm-pulls is, say, linear in the number of arms, then the MAB policy must be $\sigma$-smooth with $\sigma = O(1/n)$.

In an MAB, a strategy is an $\varepsilon$-optimal $\sigma$-smooth strategy if it is $\sigma$-smooth and there is no $\sigma$-smooth strategy with a utility that is greater by more than $\varepsilon$ (Definition 5.3). In a normal-form game, a strategy is an $\varepsilon$-approximate strong $\sigma$-smooth NE if every player's strategy is $\sigma$-smooth, and no player can unilaterally deviate to a $\sigma$-smooth strategy that improves their expected payoff by more than $\varepsilon$.

We stress that the optimal *smooth* strategy can have significantly worse expected utility compared to the best *unconstrained* policy. For example, suppose $\sigma = 1/k$ for some integer $k > 1$ and that a single arm has utility 1 whereas all others have utility 0. Here the optimal unconstrained policy has utility 1, while the optimal $\sigma$-smooth policy has utility $1/k$. This "cost of smoothness" can be quantified as $u_1 - \frac{1}{k}\sum_{i=1}^{k} u_i$, where $u_1 \geq u_2 \geq u_3 \ldots$ are the utilities of the arms sorted in decreasing order. This holds simply because the optimal $\sigma$-smooth policy assigns equal weight to the top $k$ arms, while the optimal unconstrained strategy has utility $u_1$.

The key property of smooth strategies of not putting too much mass on any particular action can be used to model unpredictable players whose strategies cannot be captured by a "small" set of actions that characterizes their actions. Such strategies can prove beneficial when predicting the move of a player can result in significant negative utility (loss) for such a player. Perhaps surprisingly, our results show that properties of games with unpredictable players allow for efficient (sublinear) verification. Daskalakis et al. [2024] study smooth strategies in games. They provide additional

---

[2]Informally, in the language of interactive proofs, the aforementioned verification procedure has the completeness property but not soundness.

motivations for studying smooth strategies, and note that a strong $\sigma$-smooth NE (where $\varepsilon = 0$) is guaranteed to exist in every normal-form game.

As noted, the study of smooth strategies in the context of normal-form games was recently proposed by Daskalakis et al. [2024]. One motivation for their study of smooth strategies comes from the area of *smoothed analysis* [Spielman and Teng, 2009] which studies the influence of *random noise* on the time complexity of algorithms. There are at least two ways of studying the smoothing influence of noise on complexity issues in games and MABs. One is to assume that the instances (e.g., the payoff matrix of the normal form game) are subjected to random noise. The other, which we focus on here, is to consider *smooth strategies* for players: namely distributions over actions that do not assign too much mass to any one action. For example, for MABs we seek strategies that maximize the expected reward, subject to the constraint that the strategy is a smooth distribution. Daskalakis et al. [2024] observe that distributions that are perturbed with random noise often have such smoothness properties. Additionally, those authors provide several further motivations for studying smooth strategies in games. They also discuss how smooth strategies can be used to generate behavior that is more similar in style to humans. Furthermore, as elaborated by Daskalakis et al. [2024] smoothens is related to several equilibrium notions in game theory.

## 2.2 Our setting

**Verification in multi-armed bandits.** In this setting, both a prover and a verifier have access to an $n$-armed bandit. This access is given via an oracle: one can query the oracle by specifying an arm, and in return receive a reward drawn from that arm's utility distribution. The prover and verifier communicate interactively, and at the end of the interaction the verifier either rejects or outputs a policy. If both the prover and the verifier follow the protocol, the verifier should output an approximately optimal smooth policy with probability at least $2/3$; this property is called *completeness*. The protocol should also satisfy *soundness*, in that even if the prover behaves arbitrarily, the verifier should output a non-optimal or non-smooth policy with probability at most $1/3$. Finally, the protocol should be efficient in that both the prover and verifier run in polynomial time, and the verifier makes $o(n)$ queries.[3]

**Verification in normal-form games.** We give a high-level overview of our results due to space constraints: details can be found in the supplementary material as well as the arXiv version Christ et al. [2025]. In this setting, both the prover and the verifier have access to a $k$-player, $n$-action normal-form game. This access is given via a game oracle: one can query the oracle by specifying a strategy profile,[4] and in return receive a vector of payoffs (one per player) corresponding to a tuple of actions drawn from the specified strategy profile. The prover and the verifier are both given an explicit description of a proposed strategy profile $\pi$, as well as an optimality parameter $\varepsilon$, a smoothness parameter $\sigma$, and a slackness parameter $\eta$. The prover and verifier communicate interactively, and at the end of the interaction the verifier either accepts or rejects. Analogously to the MAB setting, this protocol must satisfy completeness and soundness. Completeness requires that if both parties follow the protocol and $\pi$ is indeed an $\varepsilon$-approximate strong $\sigma$-smooth NE, the verifier accepts with probability at least $2/3$. Soundness requires that for any (possibly malicious and computationally unbounded) prover, if $\pi$ is not an $(\varepsilon + \eta)$-approximate strong $\sigma$-smooth NE, the verifier rejects with probability at least $2/3$. We note that adapting the MAB verification for verification in games requires certain technicalities such as introducing an additional *slackness parameter $\eta$*. Details can be found in the Games Section in the Supplementary Material.

## 2.3 Our contributions

We give a high-level overview of our results. Formal statements and proofs can be found in Section 5 and in the full version Christ et al. [2025]. We construct interactive protocols for MABs and normal-form games, where both the verifier and prover have polynomial running time in all parameters. We also present corresponding lower bounds, showing that verification using our protocols is strictly more efficient than learning in terms of query complexity, and that our protocols have near-optimal dependence on key parameters.

---

[3]As usual, it is possible to efficiently amplify the success probability from $2/3$ to $1 - \delta$ for $\delta$ arbitrarily small, both in our protocols for bandits and in the protocols for games.

[4]As in Footnote 1.

1. **Efficient verification for MABs.** We construct a protocol for verifying $\varepsilon$-optimal $\sigma$-smooth strategies where the verifier makes $\tilde{O}(\sigma n/\varepsilon^2)$ queries to the MAB oracle. The protocol consists of a single $\tilde{O}(n)$-bit message sent from the prover to the verifier. A formal statement is given in Theorem 5.5.

2. **Lower bounds for MABs.** We prove a matching lower bound of $\Omega(\sigma n/\varepsilon^2)$ on the number of queries needed by the verifier in *any* such MAB verification protocol. We also prove that learning approximately-optimal smooth strategies requires $\Omega(n)$ queries, which (together with our verification protocol) implies that verification can be more efficient than learning. A Formal statement appears in the Multi-armed bandits Section in the Supplementary Material.

3. **Lower communication using cryptography.** We show how to obtain an interactive proof for the value of the optimal smooth policy of a bandit, where the asymptotic number of arms pulls is the same as in our original protocol. The prover now sends at most $O(\lambda \cdot (n\sigma \log^3(1/\varepsilon)/\varepsilon))$ bits while affecting the probability of correctness by a negligible additive error: A constant $\lambda$ suffices to ensure an additive error of at most $10^{-6}$. Therefore, assuming $n\sigma/\varepsilon = o(n)$, we have a protocol with sublinear communication. A formal statement is given in Theorem 5.6.

4. **Efficient verification of smooth NE in games.** For normal-form games with $k$ players and $n$ actions, we construct an interactive protocol for verifying that a given strategy profile is an approximate smooth NE, with slackness $\eta > 0$. That is, the verifier accepts if the input strategy profile is an $\varepsilon$-approximate $\sigma$-smooth NE, and it rejects if the input strategy profile is not an $(\varepsilon + \eta)$-approximate $\sigma$-smooth NE. The verifier uses $\tilde{O}(k\sigma n/\eta^2)$ queries to the game oracle. The full statements and proofs appear in the Games Section in the Supplementary Material.

   In contrast, for constant $\varepsilon$, Theorem 4 in Rubinstein [2017] states a lower bound of $2^{\Omega(k)}$ queries to the game oracle for computing an $\varepsilon$-NE without the help of a prover, and this lower bound extends also to computing $\sigma$-smooth $\varepsilon$-NE for $\varepsilon$ constant and $\sigma = \Theta(1/n)$ (see Remark 12 in Daskalakis et al., 2024).[5] Thus, our verification protocol offers substantial savings in terms of $k$ as well. We also provide a linear matching lower bound based on similar ideas of the lower bound for verifying MABs.

## 2.4 Proof ideas

Our bandit verification protocol relies on the following observation. Consider a lying prover trying to convince a verifier that a given $\sigma$-smooth bandit strategy $\pi$ is approximately optimal, although there exists another $\sigma$-smooth strategy $\pi^*$ whose expected reward exceeds that of $\pi$ by more than $\varepsilon$. Consider a protocol that requires the prover to provide a good estimate of the expected utility of each arm. Because $\pi^*$ is smooth, it follows that in order to conceal the existence of $\pi^*$, the prover must lie about the utilities of *many* arms. Thus, it suffices for the verifier to independently estimate the utilities of a few randomly chosen arms, and reject if any of the prover's purported utilities are too far from the verifier's estimates. Importantly, whereas the prover must lie about *many* arms, it is enough for the verifier to catch *just one* lie. In particular, if the prover lies about an $\beta$ fraction of utilities, the verifier only needs to query roughly $1/\beta$ arms in order to detect the lie with constant probability. In more detail, assume that the verifier is promised that, if the prover is lying enough to require rejection (the lie distorts the utility of the optimal policy by at least $\varepsilon$), then the prover is in particular lying by more than $\alpha$ on at least a $\beta$ fraction of the arms for some specific known values $\alpha, \beta > 0$. This knowledge implies that $\beta n \sigma \alpha \geq \varepsilon$ and also $\alpha \geq \varepsilon$. Based on the promise, the verifier can detect the lie by pulling $O(\frac{1}{\beta} \cdot \frac{1}{\alpha^2})$ arms which by the above inequalities is at most $O(n\sigma/\varepsilon^2)$. While $\alpha, \beta$ are not known, the verifier can use a simple partitioning scheme to guess them, which adds a logarithmic term to the query complexity of the verifier.

Our lower bound of $\Omega(\sigma n/\varepsilon^2)$ on the number of queries needed for MAB verification relies on a reduction to the coin problem where one needs to decide with a few queries whether a given coin has bias $1/2 - \varepsilon$ or bias $1/2 + \varepsilon$. Our proof uses ideas from Even-Dar et al. [2002].

---

[5]Remark 12 in Daskalakis et al., 2024 refers to the case where $\varepsilon$ and $\sigma$ are both constant. However, as discussed in Footnote 8, this corresponds to $\sigma = \Theta(1/n)$ in our notation for smoothness.

Using succinct non-interactive arguments of knowledge (SNARKs) and vector commitments, we significantly reduce the communication between the prover and the verifier in our bandit verification protocol (the Preliminaries Section gives more details about these cryptographic tools). A vector commitment allows one to commit to a vector and reveal individual components whose consistency with the commitment can be proven. The commitment and opening proofs require space independent of the length of the vector. A SNARK allows a prover to succinctly prove that a given instance belongs to a polynomial-time computable relation.

A central observation in our NE verification protocol is that we can use the bandit verification protocol to ensure that no player has a profitable smooth deviation from a proposed strategy profile $\pi$. That is, for each $i \in [n]$, if all players except player $i$ behave according to the profile $\pi$, then player $i$ is choosing between $n$ actions, each of which has some fixed reward distribution.[6] This is exactly an $n$-armed bandit. So player $i$ has no profitable smooth deviation if and only if its current strategy is an optimal smooth strategy for an appropriately defined bandit. Our NE verification protocol essentially performs this bandit verification $k$ times, once for each player. We also show that a linear dependence on $k$ is necessary.

# 3 Related work

There is a substantial body of work studying algorithms for MABs for finding a distribution over actions (or a single action) that maximizes expected utility [Lattimore and Szepesvári, 2020, Garivier and Kaufmann, 2016]. Several works demonstrate a lower bound of $\Omega(n)$ (where $n$ is the number of arms) on the number of arm-pulls that are needed to find a strategy that maximizes utility for the player [Karnin et al., 2013, Chen and Li, 2015, Chen et al., 2017, Mannor and Tsitsiklis, 2004a, Even-Dar et al., 2002, Assadi and Wang, 2022]. Specifically, Even-Dar et al. [2002] provides an algorithm for identifying the best arm up to an additive error of $\varepsilon$ with probability of success at least $1 - \delta$. They achieve query complexity of $O(n \log(1/\delta)/\varepsilon^2)$ improving the naive algorithm whose query complexity is $O(n \log(n/\delta)/\varepsilon^2)$. Furthermore, they complement their result by providing a matching lower bound on the query complexity of any algorithm achieving such error guarantees in the PAC framework based on a reduction to the coin problem. A lower bound of $\Omega(n \log(1/\delta)/\varepsilon^2)$ was obtained using different methods in Mannor and Tsitsiklis [2004a]. They also show a lower bound of $\Omega((\frac{1}{\varepsilon^2})(n + \log(1/\delta)))$ for the case where the vector of utilities of the arms is known (but it is not known which arm has which utility) and the goal is to find the arm with the largest expected utility. There is also extensive work on regret minimization strategies in MABs [Lai and Robbins, 1985], a setting that we do not touch on in this paper.

A rich line of work examines the complexity of finding Nash equilibria (NE). The problem of finding an approximate NE in a normal-form game with at least 3 players is known to be complete for the complexity class PPAD [Daskalakis et al., 2009]. Even for the seemingly simpler case of 2 players, computing an exact NE is PPAD-complete [Chen and Deng, 2006], and computing an approximate NE is hard under the Exponential Time Hypothesis for PPAD [Rubinstein, 2017]. While finding a NE is hard, verifying that a given strategy profile is an (approximate) NE can be done in polynomial time.[7] Finally, the study of the query complexity of approximate NE in 2-player normal-form games has received significant attention [Babichenko, 2019, Fearnley et al., 2015, Göös and Rubinstein, 2023]. It is known [Göös and Rubinstein, 2023] that the trivial upper bound of $n^2$ is nearly tight: In certain games $\Omega(n^{2-o(1)})$ queries may be needed to find an approximate NE.

The intractability of finding NE or $\varepsilon$-NE in arbitrary normal-form games motivated Daskalakis et al. [2024] to introduce the notion of $\sigma$-smooth NE where each player places a probability mass of at most $1/(n\sigma)$ for each of the $n$ actions and $\sigma$ is a smoothness parameter in $[1/n, 1]$ (their parametrization of smoothness differs from ours: recall that we call a strategy smooth if for every action the mass on the action is at most $\sigma$). They consider two equilibrium notions related to smoothness: in a *strong $\sigma$-smooth NE*, each player is executing a smooth strategy and cannot improve their utility by deviating unilaterally to a smooth strategy. In a *weak $\sigma$-smooth NE*, again no player can improve their utility by deviating unilaterally to a smooth strategy; however, the strategies of players in a weak $\sigma$-smooth NE need not be $\sigma$-smooth. Analogous notions are defined by Daskalakis et al. [2024]

---

[6]The reward distribution for each action is fixed, because the reward distribution is a function of the strategies of the remaining players, which are fixed according to $\pi$.

[7]This is true for any problem in FNP, of which PPAD is a subset.

for $\varepsilon$-approximate (smooth) NE where players cannot improve their utilities by more than $\varepsilon$. For strong $\varepsilon$-approximate $\sigma$-smooth equilibrium, Daskalakis et al. [2024] prove there exists an algorithm for finding such an equilibrium in time $n^{O\left(k^4 \log(k/\varepsilon)/\varepsilon^2\right)}$, where $k$ is the number of players. For weak $\varepsilon$-approximate $\sigma$-smooth NE, they offer an algorithm with runtime complexity independent of $n$ (the runtime depends only on the number of players $k$, the smoothness parameter $\sigma$, and the approximation parameter $\varepsilon$).

Several works study interactive proofs for machine learning [Goldwasser et al., 2021, Mutreja and Shafer, 2023, Gur et al., 2024, Caro et al., 2024b,a]. Their main focus is on verification in supervised learning and similar settings. For instance, verifying that a proposed hypothesis satisfies the agnostic PAC requirement with respect to a (fixed and known) hypothesis class and a (fixed but unknown) population distribution, using less access (samples or queries) to the population distribution than is necessary for agnostic PAC learning. The questions of verifying MAB strategies and NE in normal-form games with few queries to the bandit or game oracle are not studied there.

# 4 Preliminaries

Let $\mathbb{N} = \{1, 2, 3, \dots\}$. For $n \in \mathbb{N}$, we denote by $[n]$ the set $\{1, 2, \dots, n\}$ and assume throughout that the set of actions of players both in MABs and games is $[n]$. For a set $\Omega$, we write $\Delta(\Omega)$ to denote the set of all probability measures defined on the measurable space $(\Omega, \mathcal{F})$, where $\mathcal{F}$ is some fixed $\sigma$-algebra that is implicitly understood. We often identify a distribution $p \in \Delta([n])$ with the vector $p = (p_1, \dots, p_n)$ such that $p_i = p(i) = \mathbb{P}_{x \sim p}[x = i]$. Finally, we define the notion of a smooth distribution:

**Definition 4.1.** *Let $n \in \mathbb{N}$ and $\sigma \in [1/n, 1]$. A probability distribution $p \in \Delta([n])$ is called $\underline{\sigma\text{-smooth}}$ if for every $i \in [n]$, $p_i \leq \sigma$.*

The degree of smoothness of a distribution is governed by the parameter $\sigma$. For $\sigma = 1$, smoothness is vacuous as every probability distribution is 1-smooth. On the other extreme, when $\sigma = 1/n$ the distribution is the smoothest possible: the uniform distribution.

# 5 Bandits

## 5.1 Definitions

**Definition 5.1** (Bandit). *Let $n \in \mathbb{N}$. An $\underline{n\text{-arm bandit}}$ is a vector of $n$ distributions $q = (q_1, \dots, q_n) \in (\Delta([0, 1]))^n$.*

*A bandit defines a $\underline{\text{bandit oracle}}$ such that, given a query $i \in [n]$ (corresponding to "pulling the $i$-th arm of the bandit"), the oracle returns a utility $x \sim q_i$ sampled independently of all previous oracle queries and responses.*

*The $\underline{\text{expected utilities vector}}$ of $q$ is a vector $u = \text{utility}(q) \in [0, 1]^n$ such that $u_i = \mathbb{E}_{x \sim q_i}[x]$ for all $i \in [n]$. A $\underline{\text{strategy}}$ for an $n$-arm bandit is a distribution $\pi = (\pi_1, \dots, \pi_n) \in \Delta([n])$. The $\underline{\text{expected}}$ $\underline{\text{utility of } \pi \text{ with respect to } u}$ is $\mathbb{E}_{i \sim \pi, x \sim q_i}[x] = \sum_{i=1}^{n} \pi_i u_i = \pi \cdot u$.*

**Definition 5.2** (Smooth bandit strategy). *Let $n \in \mathbb{N}$ and $\sigma \in [1/n, 1]$. A strategy $\pi \in \Delta([n])$ for an $n$-arm bandit is $\sigma$-smooth if $\pi_i \leq \sigma$ for all $i \in [n]$.*

**Definition 5.3** (Optimal smooth bandit strategy). *Let $n \in \mathbb{N}$, $\varepsilon \geq 0$, $\sigma \in [1/n, 1]$, let $u \in [0, 1]^n$ be the expected utilities vector of an $n$-arm bandit, and let $\pi \in \Delta([n])$ be a strategy. We say that $\pi$ is $\underline{\varepsilon\text{-competitive with respect to } \sigma\text{-smooth policies for } u}$, if for every $\sigma$-smooth strategy $\pi' \in \Delta([n])$,*

$$\pi' \cdot u - \pi \cdot u \leq \varepsilon.$$

*If in addition $\pi$ is $\sigma$-smooth, then we say that $\pi$ is an $\underline{\varepsilon\text{-optimal } \sigma\text{-smooth strategy for } u}$.[8]*

**Definition 5.4** (Verification of optimality for smooth bandit strategies). *An $\underline{\text{interactive proof system}}$ $\underline{\text{for verification of } \varepsilon\text{-optimal } \sigma\text{-smooth policies for } n\text{-arm bandits}}$ is a pair of algorithms $(V, P)$ such*

---

[8] These are special cases of definitions in Daskalakis et al. [2024]. The first definition corresponds to a weak $\varepsilon$-approximate $\sigma'$-smooth Nash equilibrium for $u$, and the second definition corresponds to a strong $\varepsilon$-approximate $\sigma'$-smooth Nash equilibrium for $u$, where $\sigma' = 1/(n\sigma)$.

*that for all $n \in \mathbb{N}$, and for every $n$-arm bandit $q$ with expected utilities vector $u = \text{utility}(q) \in [0, 1]^n$ and bandit oracle $\mathcal{O}_q$, and for all $\sigma \in [1/n, 1]$ and $\varepsilon \geq 0$, the following two conditions hold:*

- **Completeness**. *Let the random variable*
$$\pi_V = \left[V^{\mathcal{O}_q}(n, \varepsilon, \sigma), P^{\mathcal{O}_q}(n, \varepsilon, \sigma)\right] \in \Delta([n]) \cup \{\text{reject}\}$$
*denote the output of $V$ after interacting with $P$, when each of them receives $(n, \varepsilon, \sigma)$ as input and has oracle access to $\mathcal{O}_q$. Then*
$$\mathbb{P}[(\pi_V \neq \text{reject}) \wedge (\forall \, \sigma\text{-smooth } \pi' \in \Delta([n]) : \pi' \cdot u - \pi_V \cdot u \leq \varepsilon)] \geq \frac{2}{3}.$$

- **Soundness**. *For any (possibly malicious and computationally unbounded) prover $P'$ (which in particular may depend on $n$, $\varepsilon$, $\sigma$ and $q$), the verifier's output $\pi_V = \left[V^{\mathcal{O}_q}(n, \varepsilon, \sigma), P'\right] \in \Delta([n]) \cup \{\text{reject}\}$ satisfies*
$$\mathbb{P}[(\pi_V = \text{reject}) \vee (\forall \, \sigma\text{-smooth } \pi' \in \Delta([n]) : \pi' \cdot u - \pi_V \cdot u \leq \varepsilon)] \geq \frac{2}{3}.$$

*In both conditions, the probability is over the randomness of $\mathcal{O}_q$ and $V$, as well as $P$ or $P'$.*

**Theorem 5.5** (Verification for bandits). *Let $n \in \mathbb{N}$, let $\sigma \in [1/n, 1]$, let $\varepsilon \geq 0$. There exists an interactive proof system $(V, P)$ for verification of $\varepsilon$-optimal $\sigma$-smooth policies for $n$-arm bandits such that:*

- *The protocol consists of a single message of $O(n \log(1/\varepsilon))$ bits sent from $P$ to $V$.*

- *$P$ performs $m_P = O\left(n \log(n/\varepsilon)/\varepsilon^2\right)$ nonadaptive queries to the bandit oracle and runs in time $\text{poly}(n, 1/\varepsilon)$.*

- *$V$ performs*
$$m_V = O\left(\frac{n\sigma}{\varepsilon^2} \cdot \log\left(\frac{n\sigma}{\varepsilon}\right) \log\left(\frac{1}{\varepsilon}\right)\right)$$
*nonadaptive queries to the bandit oracle, and runs in time $\text{poly}(n, 1/\varepsilon)$.*

*In particular, if $\sigma = \Theta(1/\sqrt{n})$ then $m_V = \tilde{O}(\sqrt{n})$, and if $\sigma = \Theta(1/n)$ then $m_V$ is independent of $n$.*

The pseudocode of the verification protocol and the proof of Theorem 5.5 appear in the Multi-armed bandits Section in the Supplementary Material.

## 5.2 Vector commitments

**Cryptography preliminaries** Let $\lambda \in \mathbb{N}$ denote the security parameter. We write p.p.t. to mean probabilistic polynomial time. We let $\text{negl}(\lambda)$ denote a function that is $O(1/\lambda^c)$ for all $c > 0$.

A *vector commitment* [Catalano and Fiore, 2013] is a tuple of p.p.t. algorithms:

- $\text{KeyGen}(1^\lambda, n) \to \text{pp}$: takes as input the security parameter and size $n$ of the vectors to be committed, and outputs public parameters pp.
- $\text{Commit}_{\text{pp}}(v) \to c_v, \text{aux}$: takes as input a length-$n$ vector $v$, and outputs a commitment $c_v$ and auxiliary information aux. aux often contains the entire committed vector $v$.
- $\text{Open}_{\text{pp}}(v_i, i, \text{aux}) \to \text{pf}$: takes as input a value $v_i$, an index $i$, and auxiliary information aux. It outputs a proof pf that $v_i$ is the $i^{\text{th}}$ component of $v$ corresponding to aux.
- $\text{Verify}_{\text{pp}}(c_v, v_i, i, \text{pf}) \to \{\text{accept}, \text{reject}\}$: takes as input a commitment $c_v$, a value $v_i$, an index $i$, and a proof pf. It accepts if and only if $c_v$ commits to a vector whose $i^{\text{th}}$ component is $v_i$.

Vector commitments must satisfy *correctness* and *position binding*. Correctness requires that with overwhelming probability, any honestly generated public parameters and honestly committed vectors yield valid opening proofs for all of their components. Position binding requires that it is infeasible for any non-uniform p.p.t. adversary to produce a commitment and two valid proofs for *different* openings of that commitment. For precise statements, please see the Cryptography Section in the Supplementary Material.

## 5.3 Succinct non-interactive arguments of knowledge

In our protocols we use succinct non-interactive arguments of knowledge (SNARKs). More information regarding SNARKs can be found in Groth [2016].

A succinct non-interactive argument of knowledge for a relation generator $\mathcal{R}$ is a tuple of p.p.t. algorithms:

- $\mathsf{Setup}(1^\lambda, R) \to \mathsf{pp}, \tau$: takes as input the security parameter and a relation $R \in \mathcal{R}$, and outputs public parameters $\mathsf{pp}$ and a simulation trapdoor $\tau$.
- $\mathsf{Prove}(R, \mathsf{pp}, \phi, w) \to \mathsf{pf}$: takes as input a relation $R$, public parameters $\mathsf{pp}$, and a statement-witness pair $(\phi, w) \in R$. It outputs a proof $\mathsf{pf}$ of this pair's membership in the relation.
- $\mathsf{Verify}(R, \mathsf{pp}, \phi, \mathsf{pf}) \to \{\mathsf{accept}, \mathsf{reject}\}$: takes as input a relation $R$, public parameters $\mathsf{pp}$, a statement $\phi$, and a proof $\mathsf{pf}$. It should accept if and only if $\phi$ has a witness $w$ such that $(\phi, w) \in R$.

We consider SNARKs that satisfy *perfect completeness* and *computational knowledge soundness*.

Perfect completeness requires that for all $\lambda \in \mathbb{N}$, $R \in \mathcal{R}$, and $(\phi, w) \in R$:

$$\Pr_{(\mathsf{pp}, \tau) \leftarrow \mathsf{Setup}(1^\lambda, R)} [\mathsf{pf} \leftarrow \mathsf{Prove}(R, \mathsf{pp}, \phi, w) \; : \; \mathsf{accept} \leftarrow \mathsf{Verify}(R, \mathsf{pp}, \phi, \mathsf{pf})]$$

Computational knowledge soundness requires that there exists a non-uniform p.p.t. extractor that can extract a witness whenever an adversary can compute an accepting proof. That is, for all non-uniform adversaries $\mathcal{A}$, there exists a non-uniform p.p.t. extractor $\mathcal{X}_\mathcal{A}$ such that

$$\Pr \left[ \begin{array}{l} (\phi, w) \notin R \text{ and} \\ \mathsf{Verify}(R, \mathsf{pp}, \phi, \mathsf{pf}) \to \mathsf{accept} \end{array} \left| \begin{array}{l} (R, z) \leftarrow \mathcal{R}(1^\lambda) \\ (\phi, \tau) \leftarrow \mathsf{Setup}(1^\lambda) \\ ((\phi, w), \mathsf{pf}) \leftarrow (\mathcal{A} || \mathcal{X}_\mathcal{A})(R, z, \mathsf{pp}) \end{array} \right. \right] \leq \mathsf{negl}(\lambda),$$

where $(\mathcal{A} || \mathcal{X}_\mathcal{A})$ denotes that the extractor has access to the adversary's internal state and randomness.

### 5.3.1 A low-communication protocol variant

Here we outline how to use vector commitments and SNARKs in a new protocol with lower communication cost by having the prover send a commitment to the vector $\tilde{u}$ of purported rewards, rather than sending $\tilde{u}$ in full. It uses a SNARK to prove that the optimal smooth policy with respect to $\tilde{u}$ has a claimed value. The verifier then proceeds exactly as in the smooth bandit verification protocol; but instead of examining $\tilde{u}$ directly at each index, it asks the prover for the value and opening proof. Below, we sketch the ideas behind the proof of correctness. The implementation of the protocol as well as the bandit verification protocol on which the low communication variant builds on, can be found in the Bandit Section in the Supplementary Material.

**Theorem 5.6.** *Let $\lambda \in \mathbb{N}$ be the security parameter, let $n \in \mathbb{N}$, let $\varepsilon \in (0, 1)$, let $q$ be an $n$-armed bandit, and let $u$ denote be the vector of expected utilities of $q$. There exists a protocol with the following properties. The protocol consists of a trusted setup phase, in which shared parameters are generated by a trusted entity; and an interactive phase between a prover and a verifier. Assuming the security of the underlying SNARK $\Pi$ and vector commitment $\mathsf{VC}$, our protocol satisfies:*

- **Completeness:** *If the prover behaves honestly, the verifier outputs a value $t$ that is within $\varepsilon$ of the value of the optimal $\sigma$-smooth policy with probability at least $\frac{2}{3} - \mathsf{negl}(\lambda)$.*

- **Soundness:** *Even if the p.p.t. prover behaves arbitrarily, the probability that the verifier outputs a value $t$ that is not within $\varepsilon$ of the optimal value is at most $\frac{1}{3} + \mathsf{negl}(\lambda)$.*

*The efficiency of the protocol is as follows:*

- *If $\Pi$ has constant-sized proofs, and $\mathsf{VC}$ has constant-sized commitments and opening proofs, the protocol consists of $O\big(\lambda \cdot (n\sigma \log^3(1/\varepsilon)/\varepsilon)\big)$ bits sent between $P$ and $V$.*

- *$P$ performs $m_P = O\big(n \log(n/\varepsilon)/\varepsilon^2\big)$ nonadaptive queries to the bandit oracle and runs in time $\mathsf{poly}(n, 1/\varepsilon)$.*

- *V performs*

$$m_v = O\left( \frac{n\sigma}{\varepsilon^2} \cdot \log\left(\frac{n\sigma}{\varepsilon}\right) \log\left(\frac{1}{\varepsilon}\right) \right)$$

*nonadaptive queries to the bandit oracle, and runs in time* $\mathrm{poly}(n, 1/\varepsilon)$.

*Proof.* **Communication** The prover sends a commitment and SNARK proof, each of which consist of $O(\lambda)$ bits. In the interactive phase, the verifier sends $O(a_b \log(1/\epsilon)) = O(n\sigma \log^2(1/\epsilon)/\epsilon)$ indices, each of which can be written in $\log(1/\epsilon)$ bits. The prover sends $O(a_b \log(1/\epsilon))$ openings and opening proofs, requiring $O((n\sigma \log^3(1/\epsilon)/\epsilon)$ bits in total. The number of queries made by the prover and verifier to the bandit oracle is identical to in bandit verification protocol.

We prove the soundness of the protocol. Completeness is proved in the Supplementary Material.

**Soundness** Towards a contradiction, consider a p.p.t. adversary $\mathcal{A}$ that acts as the prover and with probability at least $1/3 + 1/\mathsf{poly}(\lambda)$ causes the verifier to output $t$ that is $\epsilon$-far from the true value of the optimal $\sigma$-smooth policy for some bandit $q$. Recall that computational knowledge soundness of $\Pi$ implies that there exists a p.p.t. extractor $\mathcal{X}_{\mathcal{A}}$ that, with overwhelming probability, computes $(\pi, \tilde{u})$ such that $(c_\pi, c_v, t; \pi, \tilde{u}) \in \mathcal{R}_{\mathsf{VC.pp},n}$. That is, $c_v$ is a commitment to $\tilde{u}$, $\pi$ is indeed an optimal $\sigma$-smooth policy for $\tilde{u}$, and the value of $\pi$ is indeed $t$. Now, position binding of the vector commitment implies that with overwhelming probability all openings of $\tilde{u}$ that the prover sends to the verifier are either rejected, or indeed match the corresponding component of $\tilde{u}$.

Soundness of the protocol now follows exactly from the analysis of the bandit verification protocol. □

We remark that there exist SNARKs with constant $(O(\lambda))$-sized proofs for arithmetic circuit satisfiability, which have knowledge soundness in the generic group model [Groth, 2016]. There also exist vector commitments with constant-sized commitments and opening proofs; for example, the CDH-based scheme of Catalano and Fiore [2013].

## 6 Conclusion

We have studied protocols to verify near optimality of protocols and strategies for MABs and games. A natural direction is to study interactive proofs for near optimality of policies in Markov Decision Processes Sutton and Barto [2018] with horizon larger than 1. It appears that additional ideas are needed in this setting to deal with the network structure associated with the transitions of the MDP which is absent in the MAB setting. Additionally, empirical validation of the protocols here is of interest.

Many games include multiple sequential actions of players. Such games are often represented in *extensive-form*. Devising protocols to verify properties of strategies in extensive-form is an interesting future research direction.

## Acknowledgments and Disclosure of Funding

MC was partially supported by a Google CyberNYC grant, an Amazon Research Award, and NSF grants CCF-2312242, CCF-2107187, and CCF-2212233. JS was partially supported by NSF CNS-2154149, an Amazon Research Award, and by Vinod Vaikuntanathan's Simons Investigator Award.

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
