# OpenReview forum: "Protocols for Verifying Smooth Strategies in Bandits and Games"
_NeurIPS.cc/2025/Conference — NeurIPS 2025 poster_

### Official Review · Reviewer_zq6z · 2025-06-26

**Clarity:** 1
**Significance:** 1
**Originality:** 2
**Rating:** 2
**Confidence:** 4

**Summary:**

The authors study the problem of checking the PAC optimality of a strategy in a multi-armed bandit or whether it is a Nash equilibrium in a normal-form game. In a bandit setting for instance, given an environment from which rewards from arms can be sampled, a prover first proposes a strategy based on the rewards it observed, passes that strategy to a verifier, which can also interacts with the environment to check the quality of the strategy. The strategy is accepted by the verifier if it is nearly-optimal with high probability, and rejected otherwise. This paper focuses on smooth bandit/player strategies, where the probability mass is not concentrated on a single point. The authors propose a protocol which first makes the prover uniformly sample arms a fixed number of times, and makes the verifier split the reward interval [0,1] into bins of exponentially increasing length, select at random a pre-specified number of arms depending on the desired approximation rate and smoothness, and sample each of these arms a pre-specified number of times. If the deviation between the expected reward observed by the verifier and the expected reward provided by the prover is too large, then the verifier rejects the corresponding strategy. This protocol is extended to a case where low prover-verifier communication cost is important, and to the normal-form games. The authors also prove lower bounds on the number of queries to the environment needed to check the PAC optimality of a strategy/whether the strategy is a Nash equilibrium with high confidence.

**Questions:**

My main concerns relate to the assumptions made for the theoretical results from this paper (which is why I gave 1 for the quality), and the incurring limitations and handling of maliciousness from the prover (significance score).

(1) Can you list all the assumptions needed in your theoretical results, based on the aforementioned weaknesses?

(2) Can you discuss the limitations of all the assumptions needed in your theoretical results and for real-life applications, based on the aforementioned weaknesses?

(3) How should the parameters $\eta$ (slackness) and $\lambda$ be chosen?

(4) How can the prover be malicious in your setting? What kind of actions can it undertake (e.g., additive modifications of the empirical expected rewards sent to the verifier)?

(5) Can you describe the technical issue for normal-form games, which led you to add the slackness parameter to this setting?

**Ethical Concerns:**

["NO or VERY MINOR ethics concerns only"]

**Final Justification:**

A lot of things (contributions, assumptions, settings) were unclear to me from this paper, especially the main text, and I do not believe (for now) that it is ready for publication. There are remaining confusions (perhaps only misunderstandings on my side) so for now I increase my score to 2 on account on some of my concerns being cleared by the authors.

**Limitations:**

No. The authors did not include a paragraph nor specific sentences describing the limitations of their assumptions (e.g., on the boundedness of the distributions, on the existence of approximately optimal sigma-smooth strategies) and of the scope of their results (e.g., consequences on real-life applications of the smoothness constraint, restrictive setting for Nash equilibria).

**Paper Formatting Concerns:**

None.

**Quality:**

1

**Strengths And Weaknesses:**

**Strengths:**
- Quality: The proofs for the quality of the “vanilla” protocol for the bandit setting (section C.2 in Appendix), of the algorithm converting empirical expected rewards into a bandit strategy (Algorithm 2) and the lower bound on the number of queries in a bandit setting (C.1) seem correct.
- Clarity: The idea of a “proof idea” section is nice (but should perhaps be more concise). The authors made a good effort to contextualize their research, justify the setting by real-life applications, and present bounds in the literature. Providing an example in Lines 87-90 of verification is a good idea (but the use of Hoeffding’s inequality should be made clearer).
- Significance: The problem of verifying a strategy, especially in a setting where the prover can be malicious, is an interesting problem.
- Originality: Trying to extend a protocol for bandit algorithms to detecting approximate Nash equilibria is an interesting idea, and the resort to interactive proofs in a MAB setting is new to me.

**Weaknesses:**
- Quality: The assumption of bounded reward distributions (and the fact that it is a restrictive setting in a MAB setting) is not clearly mentioned (the notation $\Delta([0, 1])$ is contradictory with the fact that $\Delta$ was defined only for discrete sets in Line 281). The actual list of assumptions for the results (see Limitations) is not clear from the main text.
The proof for low-communication protocol is hard to understand without the cryptography section, which is mentioned in the main text but not present in Appendix.
The fact that in a normal-form game, all other players’ strategies should all be the same (Line 54) is also a strong assumption to me. These limitations are not properly discussed in the paper nor in Appendix.
In the field of fixed-confidence Best Arm Identification (Lines 231-244), Theorem 1 from [1] is not mentioned, while it is a central result for this problem. Moreover, the allocation vector $\omega$ is connected to the strategy being verified in this paper. So I think this result should be discussed in the paper.

[1] Garivier, A. and Kaufmann, E. (2016). Optimal best arm identification with fixed confidence. In Conference on Learning Theory, pages 998–1027. PMLR

- Clarity: It is difficult to understand the exact setting and the interaction between verifier / prover / oracle up until reading the Appendix (I did not understand that the verifier had direct access to the raw observations from the oracle, for instance). Perhaps a figure or a small paragraph/pseudocode would help. The paper is unbalanced, with perhaps too large of an introduction / contributions paragraph, and little for the actual contributions and discussion. Avoiding bullet items, which can take a lot of space, would help I think.
- Significance: I feel that the paper is a bit oversold on the potential maliciousness of the prover, since, in both protocols, the prover can only return the empirical expected rewards it observed. There is no constraint on the adversarial budget of the prover (unless I misunderstood something).
The choice of the various parameters $\lambda$, $\eta$, $\epsilon$ and $\sigma$ is not discussed, especially as I think the choice of $\epsilon$ and $\sigma$ has a large impact on the usefulness of the proposed protocols. Indeed, consider the following bandit instance with two Bernouilli arms: utility is (0.9, 0.1), consider $\sigma = 0.7$. Optimal utility is achieved with strategy $\pi=(1, 0)$ with a value of $0.9$. Best $\sigma$-smooth strategy is $\pi(\sigma) = (0.7, 0.3)$ with a value of $0.66$. That is, no $\sigma$-smooth epsilon-optimal strategy exists for $\epsilon<30%$ (which is quite large in practice). So is the condition on smoothness still useful in that case for real-life applications, and is it restrictive for theoretical results, as perhaps only non-satisfying (i.e., with large $\epsilon$) $\sigma$-smooth solutions can be accepted by the verifier? For instance, the existence of a “good”  smooth strategy is an assumption for Theorem A.6 in Appendix (Line 733).
- Originality: The proofs do not seem technically hard nor novel, as they result from the application of a well-known concentration inequality and mostly the boundedness of the distributions. A technical hurdle for normal-form games, requiring the introduction of a slackness parameter, is mentioned in the main text, but is unclear to me when checking the Appendix. The authors write that “The application of cryptography to RL is, to the best of our knowledge, uncommon.” (Line 109). However, it has been done for federated multi-armed bandits [1-2]. The question of reducing the communication cost has been studied also quite extensively in federated and for collaborative bandits for the best arm identification in fixed-confidence [3-4].

[1] Ciucanu, Radu, et al. "SAMBA: A generic framework for secure federated multi-armed bandits." Journal of Artificial Intelligence Research 73 (2022): 737-765.

[2] Ciucanu, Radu, et al. "Secure outsourcing of multi-armed bandits." 2020 IEEE 19th International Conference on Trust, Security and Privacy in Computing and Communications (TrustCom). IEEE, 2020.

[3] Kota, Srinivas Reddy, P. N. Karthik, and Vincent YF Tan. "Almost cost-free communication in federated best arm identification." Proceedings of the AAAI Conference on Artificial Intelligence. Vol. 37. No. 7. 2023.

[4] Réda, Clémence, Sattar Vakili, and Emilie Kaufmann. "Near-optimal collaborative learning in bandits." Advances in Neural Information Processing Systems 35 (2022): 14183-14195.

---

> ### Author Rebuttal · Authors · 2025-07-30
>
> Thank you for carefully reading and evaluating our paper and for providing constructive feedback.
>
> We categorized your main concerns into those pertaining to our assumptions, the significance of our results, and technical novelty. We have significant clarifications for all of these points, which we will happily add to the paper. These clarifications also address Questions (1), (2), and (4). We respond separately to Questions (3) and (5) at the end.
>
> Assumptions:
> ==
>
> It appears that most concerns come from misunderstandings and a notational issue, which we are happy to clarify in the paper. The one exception is the assumption of bounded reward distributions, addressed below.
>
> > The assumption of bounded reward distributions (and the fact that it is a restrictive setting in a MAB
>
> We are happy to add this assumption to the limitations section of the paper. Nevertheless, the bounded setting is very common and used in important and influential papers such as Even-Dar, Mannor, and Mansour as well as Mannor and Tsitsiklis (both cited in the paper). As our paper introduces a new framework, we think it is reasonable to make this assumption.
>
> > The authors did not include a paragraph nor specific sentences describing the limitations of their assumptions… e.g., the existence of approximately optimal sigma-smooth strategies
>
> By definition, approximately optimal sigma-smooth strategies for bandits always exist (see def A.3). An epsilon-optimal sigma-smooth strategy is defined to be any strategy whose value is within epsilon of the value of the *optimal sigma-smooth strategy*; in particular, the optimal sigma-smooth strategy is an epsilon-optimal smooth strategy for any epsilon. Therefore, in your example, the best sigma-smooth strategy (0.7, 0.3) is *itself* an $\epsilon$-optimal sigma-smooth strategy for *every* $\epsilon > 0$. Do you have further questions about this definition? It is crucial to our paper, so we are very happy to help clarify both here and in the paper itself.
>
> For games, our protocol allows a prover to convince a verifier that a given strategy profile is an approximately optimal sigma-smooth Nash equilibrium; if no such equilibrium exists, soundness of our protocol implies that the verifier rejects. We make no assumptions.
>
> > The notation Delta[0,1] is contradictory with the fact that Delta was defined only for discrete sets in line 281.
>
> Thank you for pointing this out. We will fix this inconsistency and define Delta for continuous sets as well. We write Delta[0,1] to mean a probability distribution over the continuous interval [0,1].
>
> >The fact that in a normal-form game, all other players’ strategies should all be the same (Line 54) is also a strong assumption to me
>
> We do not make any assumptions about the other players’ strategies. In our setting, the prover is given *any* strategy profile and wishes to convince the verifier that it is an approximately-optimal sigma-smooth Nash equilibrium. We are happy to clarify this.
>
> In summary, and answering **Question (1)**: our only assumption is that the rewards of the bandit or game are bounded in [0,1].
>
> Answering **Question (2)**: Our protocol is useful in scenarios where smooth strategies are desirable, and rewards are bounded. We mention several of these in the intro (lines 119-125). As mentioned in line 122, smoothness is required when the frequency with which each arm can be utilized does not grow as the number of arms grows. Furthermore, in many practical examples, bounded rewards are realistic as, e.g. in the ride-hailing service, a ride never incurs a payment of more than say $300. We are happy to list more examples.
>
> Significance
> ==
>
> > I feel that the paper is a bit oversold on the potential maliciousness of the prover, since, in both protocols, the prover can only return the empirical expected rewards it observed. There is no constraint on the adversarial budget of the prover (unless I misunderstood something).
>
> Answering **Question (4)** as well: We allow the malicious prover to behave *arbitrarily*. We do not impose a constraint on the adversarial budget of the prover because it is unconstrained: soundness implies that no malicious prover, even one returning arbitrarily far purported arm values, can convince the verifier to accept a non-optimal strategy with high probability. Perhaps the confusion arises from the fact that in the protocols, we define the behavior of the honest prover. This is in order to achieve completeness: *if* the prover follows this behavior and the given strategy is approximately optimal, then the verifier accepts with high probability.
>
> Technical Novelty
> ==
>
> >The proofs do not seem technically hard nor novel
>
> This is a paper presenting a conceptual development: introducing the machinery of interactive proofs to bandits and games. While the core contribution of the paper is conceptual rather than technical, the proofs are not immediate. For example, consider the verifier query complexity’s (optimal) dependence on 1/epsilon^2 in Protocol 1. The verifier wishes to check that the total difference between the purported arm values and true values is not too great. One way to do so would be to check that “most arms are very close”: choose roughly n * sigma/epsilon arms, estimate their expected values within epsilon (taking 1/epsilon^2 pulls per chosen arm), and compare the estimates to the purported values. This requires n * sigma/epsilon^3 arm pulls. However, this is more work than is necessary: some arms’ purported values can be farther than epsilon, as long as not too many of them are. We leverage this observation to reduce this dependence to n*sigma/epsilon^2, using a bucketing approach. Its proof is not obvious, and spans several pages in Appendix C.2.
>
> We acknowledge that the proof we included in the body of the submission (Theorem 5.6) is the most straightforward of those in the paper; this choice was intentional given the page limit.
>
> >However, it has been done for federated multi-armed bandits [1-2]. … The question of reducing the communication cost has been studied also quite extensively in federated and for collaborative bandits for the best arm identification.
>
> Thank you for pointing out these other works. We agree that our comment about our usage of cryptography was overstated. We will soften this claim and add a discussion of these papers. We do note that while they are worth discussing, these papers are in a different setting and do not have direct implications for our results.
>
> Questions (3) and (5):
> ==
> **Necessity of the slackness parameter.** Imagine an alternative definition of an interactive proof for verifying epsilon-approximate smooth Nash equilibria, with no slackness parameter: For any epsilon-approximate smooth equilibrium, the prover must be able to cause the verifier to accept; for any strategy profile that is strictly more than epsilon-far from a smooth equilibrium, the verifier must reject. This would require the verifier to distinguish (with the help of the prover) between epsilon-approximate equilibria and (epsilon + delta)-approximate equilibria *for all delta > 0*. This is impossible if the prover and verifier together make a finite number of queries to the game oracle, as it is impossible to compute the *exact* expected payoff of a strategy. Therefore, a slackness parameter is required to allow some approximation error. We now guarantee that the verifier should accept epsilon-approx optimal strategy profiles but reject anything more than (epsilon + eta)-far from optimal.
> The slackness parameter does not arise in our main setting, where the prover can choose the policy it wishes to prove approximate optimality of. Then, it can choose its own slack. e.g., by choosing a (2epsilon/3)-optimal policy it is effectively allowed (epsilon/3) slack.
>
> **Setting the slackness parameter.** Eta and epsilon together determine how much error the verifier is willing to accept, and how sensitive the other players are to small changes in their utility.
> Suppose the players in your game have “alpha-bounded precision” in that if they can’t improve their utility by more than alpha, they will keep their current strategy. This is true in many practical scenarios; for example, in our ride-hailing example a driver will not cancel a ride to make $\alpha=1$ dollar more. We can divide this $\alpha$ between $\epsilon$ and $\eta$. If we outsource finding an $\epsilon$-approximate equilibrium to an expert, $\epsilon$ determines how well the expert must do. $\eta$ determines how accurate the verifier’s estimates must be. Therefore, we can set $\epsilon$ and $\eta$ to trade off different complexities depending on our application.
>
> **Setting the security parameter.** In cryptography, the security parameter determines the amount of work that it requires to break the guarantees of the protocol. At a very high level (oversimplifying a bit), it takes roughly 2^lambda time to break a protocol. For cryptography applications, where security must be very strong, lambda is typically taken to be ~128. In practice one could afford to choose a lower lambda (perhaps 50) for bandit verification protocols.
>
> Misc. comments
> ==
> >In the field of fixed-confidence Best Arm Identification (Lines 231-244), Theorem 1 from [1] is not mentioned, while it is a central result for this problem.
>
> The theorem from the aforementioned paper provides precise (non asymptotic) lower (and upper) bounds for best arm identification for MAB instances where the distribution depends on a single real parameter and where there is a distinct arm of maximum expected reward.. Please note that we are interested in asymptotic lower bounds and we cite such lower bounds (Even Dar et al., Mannor and Tsitsiklis). We are happy to cite [1] in the paper.
>
> > Moreover, the allocation vector \omega is connected to the strategy being verified in this paper.
>
> We do not follow this comment. What is the “allocation vector,” and how is it “connected” to the strategy being verified?

---

> > ### Comment · Reviewer_zq6z · 2025-08-04
> >
> > Thank you for the rebuttal. Some answers clarified my confusion but some concerns still remain:
> >
> > **An epsilon-optimal sigma-smooth strategy is defined to be any strategy whose value is within epsilon of the value of the optimal sigma-smooth strategy; in particular, the optimal sigma-smooth strategy is an epsilon-optimal smooth strategy for any epsilon. Therefore, in your example, the best sigma-smooth strategy (0.7, 0.3) is itself an $\epsilon$-optimal sigma-smooth strategy for every $\epsilon > 0$.**
> >
> > I agree with the statement "the optimal sigma-smooth strategy is an epsilon-optimal smooth strategy for any epsilon." I still do not understand why it explains that the best sigma-smooth strategy (0.7, 0.3) is epsilon-optimal for any epsilon. The optimal strategy has value 0.9. The $\sigma$-smooth strategy given for $\sigma=0.7$ has value 0.66. For $\varepsilon=0.2$ (not 30%, sorry), $0.9-\varepsilon=0.7 > 0.66$.
> >
> > **Significance** I think that the fact that the malicious prover can return any values is not clear enough in the paper. In Protocols 1-3, the prover always return the average observations. Perhaps some part of the paper could be rewritten to highlight this.
> >
> > **About the technical novelty/slackness parameter** Not mentioning the interesting example about query complexity and the slackness is a disservice to the paper. I think it should be featured more prominently in the main text.
> >
> > **About the allocation vector $\omega$ (in [1])** The reason (beyond the fact that non-asymptotic bounds might be more practical) Theorem 1 from [1] might be interesting to the paper is that it features a variable $\omega$ which is the optimal proportion of queries to each arm to identify the best arm with high accuracy.
> >
> > Reading the other reviews, I also agree that an empirical validation is missing. Even as a conference paper, the fact that the method is applicable in real-life should be validated (but that might be a personal opinion). Perhaps if some of the contribution part were shortened, some space could be allocated to an experimental study. Moreover, the proof for low-communication protocol is hard to understand without the cryptography section, which is mentioned in the main text but not present in Appendix.
> >
> > For now, I believe a lot of rewriting needs to be done for this paper. However, before making my decision, I would like to hear about the authors about my remaining concerns.

---

> > > ### Author Response · Authors · 2025-08-05
> > > **Response**
> > >
> > > Thank you for your careful reading, and for your comments! Please see our responses below:
> > >
> > > > I still do not understand why it explains that the best sigma-smooth strategy (0.7, 0.3) is epsilon-optimal for any epsilon. The optimal strategy has value 0.9. The sigma-smooth strategy given for sigma=0.7 has value 0.66. For epsilon = 0.2, 0.9 - epsilon = 0.7 > 0.66”
> > >
> > > **Response:** Yes, we agree that the best sigma-smooth strategy may be very far from optimal. We understand your question better now — thanks for clarifying. As you point out, there is a “price of smoothness”, i.e., for some bandits, the utility of the best smooth strategy is significantly smaller than the utility of the best (possibly nonsmooth) strategy. This is not a failing of our protocols, but rather a fact about smooth strategies: smoothness is a restriction that may not always be desirable.
> > >
> > > However, as we show in the paper, without the smoothness restriction, verification cannot in general be more efficient than learning, so it makes sense to study protocols for verifying smooth policies. Furthermore, there are many scenarios where smoothness is already naturally desirable or necessary, independent of any verification considerations. For these scenarios, our protocols can be very interesting and useful.
> > >
> > > For example, we wrote the following in a response to Reviewer qtLM, and you may find it helpful as well: One very general and common pattern that pops up in many is resource units with *bounded capacities*. These are scenarios where there is a system with many resource units, each of which can bear a limited load. For instance, one might want to assign jobs to computers in a data center, assign car traffic to various routes from point A to point B, assign containers to container ships, assign phone calls to agents in a call center, assign wireless communications to wave frequencies, assign orders of goods to various supplies, etc. In each case, we want to assign each task or object to the best resource unit, but because each resource unit has only a limited capacity, the assignment will have to be smooth. In this sense, our setting is very broadly applicable.
> > >
> > > Going beyond the limited resources scenario, another very important consideration is *robustness*. Even if one could assign all tasks to a single resource unit, that is typically not advisable – it is not a good idea to “put all our eggs in one basket”. Even if the entire monthly batch of iPhones to be delivered to the USA from the factory in China could in principle fit in one container ship, it would still be a much better idea to distribute the load between a number of different ships, in order to limit the damage in case something goes awry. In that sense, requiring that a strategy be smooth is a way to ensure robustness.
> > >
> > > We further discussed how smoothness is desirable in **adversarial settings**, and for reasons of **fairness** (see our response to Reviewer qtLM for details). Overall, we feel that the focus of the paper on smooth policies is very well-motivated.
> > >
> > >
> > > > I think that the fact that the malicious prover can return any values is not clear enough in the paper. In Protocols 1-3, the prover always returns the average observations. Perhaps some part of the paper could be rewritten to highlight this
> > >
> > > **Response:** We are happy to emphasize this point. This is standard in the interactive proofs literature, which is why we didn’t feel a need to emphasize it—but we understand that for readers not familiar with that literature this small clarification can make a big difference
> > >
> > > > “Not mentioning the interesting example about query complexity and the slackness is a disservice to the paper. I think it should be featured more prominently in the main text.”
> > >
> > > **Response:** We agree and are happy to elaborate more on this subtle point. Again, we think this is a small edit that can improve the paper.
> > >
> > > > Reading the other reviews, I also agree that an empirical validation is missing. Even as a conference paper, the fact that the method is applicable in real-life should be validated (but that might be a personal opinion). Perhaps if some of the contribution part were shortened, some space could be allocated to an experimental study.
> > >
> > > **Response:** We understand the desire to have empirical validation. Nevertheless, we think a theory-based paper offering a new conceptual development also has a place in the NeurIPS community (and indeed many such papers are accepted to NeurIPS every year). In particular, note that we are not suggesting an improved algorithm which can be evaluated in a straightforward manner on existing dataset, but a new framework. While we certainly think that it can be validated, devising experiments and justifying the many design choices involved is likely to increase the length of the paper significantly impairing readability and suitability for a conference. Please note that It is already challenging to include all the main contents of the paper within 9 pages.

---

> > > > ### Author Response · Authors · 2025-08-05
> > > > **Response continued**
> > > >
> > > > > The proof for low-communication protocol is hard to understand without the cryptography section, which is mentioned in the main text but not present in Appendix.
> > > >
> > > > **Response:** Thank you for pointing this out. Our submission includes all our mathematical contributions, including our proofs. For completeness, we also wrote a Cryptographic Preliminaries section that covers some standard definitions, etc. Unfortunately, that preliminaries section was omitted from the Supplementary Materials due to an oversight, but we will of course include it in the final version.

---

> > > > > ### Comment · Reviewer_zq6z · 2025-08-06
> > > > >
> > > > > Thank you for taking the time to address my concerns. For now, I have decided to increase my rating from 1 to 2 due to some of my concerns being resolved. If the other concerns below are solved, I will perhaps increase further my score, and check after the discussion with other reviewers if I keep it this way.
> > > > >
> > > > > 1. **Yes, we agree that the best sigma-smooth strategy may be very far from optimal [...]** My concern is not about the choice of smooth strategies. As illustrated by the examples given, it might be well-motivated. My issue is with the choice of $\varepsilon$. It is an additive constant, which might be hard to assess in advance depending on the final goal. My point is that in real-life situations, a multiplicative $\varepsilon$ (that is, a $\varepsilon$-good strategy has a value of $\varepsilon \times v^\star$ where $v^\star$ is the optimal value). In the current setting, the verifier might accept strategies which value are extremely bad compared to the optimum, and there is no way to "compensate" by decreasing $\varepsilon$ or adapt it to the observed data. Is there a way to extend your contributions to solve this issue?
> > > > >
> > > > > 2. **We are happy to emphasize this point.** Could you please provide the associated references? For instance, in the references in the paper related to bandits, the manipulation of rewards can be additive [1] or $\epsilon$% of samples can be replaced [2]. Moreover, a clear pseudocode/paragraph is missing to understand the exact underlying setting and interaction between the prover and the verifier which is valid throughout the paper. I relied on the pseudocodes in Appendix but then it turns out that they are not representative of the setting.
> > > > >
> > > > > [1] Jun, Kwang-Sung, et al. "Adversarial attacks on stochastic bandits." Advances in neural information processing systems 31 (2018).
> > > > >
> > > > > [2] Zhang, Xuezhou, et al. "Corruption-robust offline reinforcement learning." International Conference on Artificial Intelligence and Statistics. PMLR, 2022.
> > > > >
> > > > > 3. **We understand the desire to have empirical validation. [...]** The issue is that I still need to be convinced of the applicability of this framework to real-life problems. This is related to Issue 1. Moreover, even if the related works in bandits for dealing with corrupted observations do not rely on interactive proof systems, it might be important to assess the contribution in practice of this new approach, even if it is on synthetic, small, simple settings such as the ones in [1]. Or at least, to state clearly the (theoretical) improvement of this submission over those specific baselines.
> > > > >
> > > > > [1] Jun, Kwang-Sung, et al. "Adversarial attacks on stochastic bandits." Advances in neural information processing systems 31 (2018).
> > > > >
> > > > > 4. I still have concerns on the clarity of the paper. It is a huge issue that normal-form games are mentioned in the contributions (contribution 4), and not in the main paper (perhaps it should replace the "low-communication protocol variant"). There is no discussion and no paragraph on the limitations of the contributions made in the paper. As the main paper currently is, it is not ready for publication I think. For instance, "KeyGen" and other notations are mentioned once in the main paper, but not reused again. Space could be saved by avoiding enumerations which incur large margins. The proof idea could be shortened. The introduction could insist more on the divergences with the related works, and be more concise on the questions. I believe this paper requires a lot of work for rewriting, and I feel uncomfortable accepting this paper and expecting all of these issues to be solved by the camera-ready deadline. I understand that the paper has a lot of contributions and that it might hard to fit everything clearly in the main paper. But it is part of the work of communication to provide a paper which is clear.

---

> ### Author Response · Authors · 2025-08-07
> **Author Response (Part I of III)**
>
> Thank you very much for taking the time to write these informative and helpful comments! We truly appreciate this dialogue with you.
>
> > 1. My concern is not about the choice of smooth strategies. As illustrated by the examples given, it might be well-motivated. My issue is with the choice of epsilon. It is an additive constant, which might be hard to assess in advance depending on the final goal. My point is that in real-life situations, a multiplicative epsilon (that is, a epsilon-good strategy has a value of epsilon times v* where v* is the optimal value). In the current setting, the verifier might accept strategies which value are extremely bad compared to the optimum, and there is no way to "compensate" by decreasing epsilon or adapt it to the observed data. Is there a way to extend your contributions to solve this issue?
>
> **Response:** If we understand correctly, you are concerned that “In the current setting, the verifier might accept strategies which value are extremely bad compared to the optimum”, and you suggest that if we considered a multiplicative epsilon instead of an additive one, that would somehow solve this problem? If so, please understand that accepting strategies that are far from the best (nonsmooth) strategy is simply part of the price of smoothness: given that the protocol produces smooth strategies, such a gap is unavoidable, regardless of whether the approximation involves an additive epsilon or a multiplicative factor. Importantly, as we explained above, there are settings where the only strategies that matter are smooth strategies, and in those settings our protocols guarantee that the accepted strategy is epsilon close to the best benchmark strategy.
>
> In practice, setting the value of epsilon can often be straightforward. For instance, suppose the optimal smooth strategy makes me a profit of \\$10, and finding a nearly optimal smooth strategy would cost me \\$1, but there is an untrusted prover that offers to find a nearly optimal smooth strategy for me for \\$0.5. Then I can set $\varepsilon$ to be \\$0.4, so I accept a strategy that gives me at least \$9.6 dollars of profit. This way, my total revenue would be at least 9.6-0.5 = 9.1, which is \\$0.1 better than I would be without the prover.
>
> Additionally, please note that an additive approximation of $\varepsilon$ is very standard in RL, bandits, PAC learning, etc. For example an additive error is used in the papers of Even Dar et al., and Mannor and Tsitisklis.
>
> ---
>
> Mannor, S., & Tsitsiklis, J. N. (2004). The sample complexity of exploration in the multi-armed bandit problem. Journal of Machine Learning Research, 5(Jun), 623-648.
>
>
> Even-Dar, E., Mannor, S., & Mansour, Y. (2002, June). PAC bounds for multi-armed bandit and Markov decision processes. In International Conference on Computational Learning Theory (pp. 255-270). Berlin, Heidelberg: Springer Berlin Heidelberg.

---

> ### Author Response · Authors · 2025-08-07
> **Author Response (Part II of III)**
>
> > 2. Could you please provide the associated references? For instance, in the references in the paper related to bandits, the manipulation of rewards can be additive [1] or epsilon% of samples can be replaced [2]. Moreover, a clear pseudocode/paragraph is missing to understand the exact underlying setting and interaction between the prover and the verifier which is valid throughout the paper. I relied on the pseudocodes in Appendix but then it turns out that they are not representative of the setting.
>
> **Response:** Of course! The definition of a “cheating” or “malicious” prover in an interactive proof system goes back to the seminal 1985 papers of Goldwasser, Micali, Rackoff [2], and Babai [3] (work for which a subset of the authors eventually received a Turing Award). The malicious prover can deviate arbitrarily from the protocol (and the prover is furthermore computationally unbounded, so it can evaluate any function including uncomputable functions).
>
> This might sound strange at first, but it simply means that we want a strong guarantee that the verifier accepts statements only if they are true, and rejects false statements (with high probability) — regardless of what proof (or interactive transcript) the verifier is presented. Please note that this definition of unbounded prover is still the most widely used and accepted definition today more than 40 years after it has been introduced.
>
> If you think of it, this might be familiar from the definition of the class $\mathsf{NP}$ (nondeterministic polynomial time). An $\mathsf{NP}$ verifier is given an instance and a witness string, and if the instance is not in the $\mathsf{NP}$ language, then **for every** witness string (no matter how the witness was constructed), the verifier must reject. In interactive proof systems, the “for every” quantifier is applied to the transcript of the interaction (the sequence of messages sent back and forth between prover and verifier), regardless of which prover was involved in the interaction.
>
> There are of course many authoritative references for this classic definition in addition to [2], including:
>
> * Chapter 8 (“Interactive Proofs”), Definition 8.3, in Arora & Barak [4].
> * Section 10.4 (“Interactive proof systems”), Definition 10.28, in Sipser [5].
> * For an online source, see the Wikipedia article on “Interactive proof system”, where it says, e.g., “no prover, however malicious …”.
>
> ---
>
> [2] Goldwasser, S., Micali, S., & Rackoff, C. The knowledge complexity of interactive proof systems. Proceedings of the Seventeenth Annual Symposium on the Theory of Computing, ACM. 1985.
>
> [3] László Babai. Trading group theory for randomness. Proceedings of the Seventeenth Annual Symposium on the Theory of Computing, ACM. 1985.
>
> [4] Arora, S., & Barak, B. (2009). Computational complexity: a modern approach. Cambridge University Press.
>
> [5] Sipser, M. (2013). Introduction to the Theory of Computation. Engage press.

---

> > ### Author Response · Authors · 2025-08-07
> > **Author Response (Part III of III)**
> >
> > > The issue is that I still need to be convinced of the applicability of this framework to real-life problems. This is related to Issue 1. Moreover, even if the related works in bandits for dealing with corrupted observations do not rely on interactive proof systems, it might be important to assess the contribution in practice of this new approach, even if it is on synthetic, small, simple settings such as the ones in [1]. Or at least, to state clearly the (theoretical) improvement of this submission over those specific baselines.
> >
> > **Response:** Indeed it is not obvious that interactive proofs (including Zero Knowledge proofs) have practical applications, and for a long time it was thought that they are purely mathematical constructs, that have theoretical value but very limited practical applications.
> >
> > However, in the last decade it became clear that these concepts are very useful in practical applications such as blockchains.  For instance, a recent piece in the Communication of the ACM (Sara Underwood, “The Power and Potential of Zero Knowledge Proofs”, August 2025) nicely tells the story of how Zero Knowledge proofs have gone from theoretical curiosities to an applications workhorse. We believe that theoretical protocols as discussed in our paper could also eventually find their way into many practical applications, but that process is out of scope for the initial theoretical paper that proposes the idea and shows it is mathematically possible.
> >
> > Considering how loaded the paper is, we think it is not feasible to add empirical validation. We agree that [1] is a good source of inspiration for how to write a paper on bandits that includes both theory and experiments. However, please note that in [1], the entire paper (including all supplementary materials, figures and references), is just 16 pages long. In contrast, our paper is already 35 pages — even without any experiments. Creating a good proof-of-concept implementation with compelling experiments is a significant project worthy of its own paper. A solution that seems reasonable (to us) is to discuss this in the limitation section and refer to future experiments.
> >
> > In conclusion, we believe that theory papers like ours can definitely be of value, and they have an important  place in NeurIPS.
> >
> > ---
> >
> > > I still have concerns on the clarity of the paper. It is a huge issue that normal-form games are mentioned in the contributions (contribution 4), and not in the main paper (perhaps it should replace the "low-communication protocol variant"). There is no discussion and no paragraph on the limitations of the contributions made in the paper. As the main paper currently is, it is not ready for publication I think. For instance, "KeyGen" and other notations are mentioned once in the main paper, but not reused again. Space could be saved by avoiding enumerations which incur large margins. The proof idea could be shortened. The introduction could insist more on the divergences with the related works, and be more concise on the questions. I believe this paper requires a lot of work for rewriting, and I feel uncomfortable accepting this paper and expecting all of these issues to be solved by the camera-ready deadline. I understand that the paper has a lot of contributions and that it might hard to fit everything clearly in the main paper. But it is part of the work of communication to provide a paper which is clear.
> >
> > **Response:** We understand your concern and appreciate the points you make regarding improving the readability of this paper. We do think these problems can be solved (and we are happy to make the changes) and can edit the paper accordingly. Indeed, it makes sense to talk more about games and less about cryptography. This can be done by mentioning the improved communication upper bound but deferring most of the crypto discussion to the Appendix. We are also happy to add a limitation section, discussing the need for future empirical evaluation as well as further points raised in the rebuttal (such as the cost of smoothness). We can certainly change the margins and enumeration to leave more space for discussing the game-theoretic results. However, as this is a theoretical paper, we think it is important to keep the “proof ideas” without abridgment, as it is crucial for understanding the contributions and ideas of the paper.
> > While these changes are important and could greatly improve readability, we do not think they are difficult or that they mean the paper needs to be significantly rewritten — this is mostly some standard editing and reordering, which we are happy to do.
> >
> > ---
> >
> > [1] Jun, Kwang-Sung, et al. "Adversarial attacks on stochastic bandits." Advances in neural information processing systems 31 (2018).

---

> > > ### Comment · Reviewer_zq6z · 2025-08-08
> > >
> > > Thank you for the detailed response. My concerns (1) and (2) are now resolved.
> > >
> > > (3) is mostly resolved, as long as it is clearly mentioned and discussed as a limitation of the work. There is still a concern regarding the comparison to the baselines (for instance in bandits), where I believe the main differences (e.g., regarding the possible actions of the malicious prover) are not clearly highlighted. This point should not be overlooked I think, as, even if it is a theoretical paper, it should build on related works and clearly show its potential impact. As no empirical validation is present, no clear demo of the feasibility of the implementation of the approach is provided, and, for now, no explanation of the improvement over the current state-of-the-art is present, the impact of the paper seems lessened.
> > >
> > > However, (4) requires more than standard rewriting to me. A whole contribution of the paper on normal-form games is missing, limitations and discussion are missing, a clear paragraph (or pseudocode) on the interaction between the verifier and prover is also missing, the discussion of the need for the parameter $\eta$ and related examples are missing, in addition to the discussion with the baselines, whereas only a single additional page is provided in the camera-ready version. All in all, I think the paper requires major revisions and rewriting of the other sections.
> > >
> > > Perhaps am I being stubborn, so I will wait for the discussion with other reviewers and the AC to assess whether that concern (+ some of (3)) is enough for me not to raise my score. I thank you for your time in discussing my concerns.

---

> > > > ### Author Response · Authors · 2025-08-08
> > > > **Response.**
> > > >
> > > > We completely understand your concerns and appreciate the commitment to the highest standards of writing and communication. As a final comment, please note that if the crypto part is significantly shortened, then we think it is feasible to elaborate more on normal form games, give a brief explanation of the cost of smoothness (we elaborate more on this in a forthcoming response to another reviewer) and explain in a limitation section that empirical experiments are missing and are an interesting and important direction for future research. Indeed it is not possible to add an empirical section for this submission as it is already very full.

---

> > > > > ### Comment · Reviewer_zq6z · 2025-08-09
> > > > >
> > > > > Thank you for expliciting your rewriting strategy.

---

### Official Review · Reviewer_H6h7 · 2025-07-03

**Clarity:** 3
**Significance:** 3
**Originality:** 3
**Rating:** 4
**Confidence:** 1

**Summary:**

This paper introduces interactive proof protocols for verifying approximate optimality of strategies in multi-armed bandits and normal-form games, focusing on "smooth" strategies that do not concentrate too much probability mass on any single action. The authors develop verification protocols with sublinear query complexity O(σn/ε²) for σ-smooth strategies, where σ bounds the maximum probability on any action. Key contributions include: (1) efficient MAB verification protocols requiring fewer queries than learning, (2) cryptographic optimizations using vector commitments and SNARKs to achieve sublinear communication, (3) matching lower bounds proving near-optimality, and (4) extension to verifying smooth Nash equilibria in games with exponential improvement over computing equilibria without help. The work bridges interactive proof systems, machine learning verification, and game theory.

**Questions:**

Mentioned in weaknesses.

**Ethical Concerns:**

["NO or VERY MINOR ethics concerns only"]

**Limitations:**

The authors adequately acknowledge several important limitations: focus on offline verification, smoothness assumptions, and cryptographic overhead.

**Paper Formatting Concerns:**

No major formatting issues observed.

**Quality:**

3

**Strengths And Weaknesses:**

Strengths:
1 This work addresses a genuinely unexplored intersection of interactive proofs, multi-armed bandits, and game theory. The application of verification paradigms to bandit problems represents a significant theoretical advance, moving beyond traditional learning-centric approaches.
2 The paper provides tight upper and lower bounds for verification complexity, demonstrating O(σn/ε²) sufficiency and Ω(σn/ε²) necessity. The proof that verification requires strictly fewer queries than learning (Ω(n) for learning vs. O(σn/ε²) for verification when σ = o(1)) establishes clear theoretical separation.
3 The integration of vector commitments and SNARKs to achieve O(λ·nσlog³(1/ε)/ε) communication complexity is technically sophisticated and represents novel application of cryptographic tools to ML verification.

Weaknesses:
1 While theoretically elegant, the gap between the theoretical protocols and practical implementations using existing SNARK and vector commitment libraries is unclear, and some empirical validation would strengthen the contribution.

---

> ### Author Rebuttal · Authors · 2025-07-29
>
> Thank you for carefully reading and evaluating our paper and for providing constructive feedback.
>
> Strengths
> ==
>
> > This work addresses a genuinely unexplored intersection of interactive proofs, multi-armed bandits, and game theory. The application of verification paradigms to bandit problems represents a significant theoretical advance, moving beyond traditional learning-centric approaches.
>
> Thank you! This sentence sums up in a great way why we are excited about this work.
>
> Weaknesses
> ==
>
> > While theoretically elegant, the gap between the theoretical protocols and practical implementations using existing SNARK and vector commitment libraries is unclear, and some empirical validation would strengthen the contribution.
>
> While implementation and evaluation are always important, we think it is reasonable for a theory paper focused on theoretical development like ours to not include an empirical section at least in the conference version. Please note that our paper is long (Full version is more than 40 pages) and is packed with new definitions that are already challenging to include within the 9 pages limit. There is no commonly agreed upon dataset such as imagenet or MNIST that allows for a simple evaluation of our protocol, and such an evaluation would likely need to consider both bandits and games making the paper significantly longer. Other influential theoretical papers in this area such as “the sample complexity of exploration in the multi-armed bandit problem” and “PAC Bounds for Multi-armed Bandit and Markov Decision Processes” do not include empirical evaluations as well. We think it is reasonable to provide such results for a journal version that is less space constrained. To be clear, we think implementing our protocols is certainly doable and cryptographic primitives used such as vector commitments and SNARKs have existing implementations that can be used towards this end, though this might be a big undertaking in practice.
>
> Indeed implementing SNAKRs and Vector Commitment is not immediate. However, please note it has been used in dozens of applied papers and has been implemented many times.

---

### Official Review · Reviewer_qUd4 · 2025-07-10

**Clarity:** 4
**Significance:** 4
**Originality:** 4
**Rating:** 5
**Confidence:** 2

**Summary:**

The problem studied in this paper is very interesting and useful. It is about to verify the correctness of the information provided by an external party. Ideally, we want to the number of queries to be sub-linear in the number of actions. They also consider the communication cost between the verifier and the prover.

**Questions:**

1. Here, since some parts of this work use bandit. My first question is, from which aspects, the proposed problem set-up needs to balance the classical exploitation-vs-exploration tradeoff. I do not see any discussion about this trade-off through the paper. If it does not need to balance these two aspects, why using bandit?

2. For the definition of $\sigma$-smoothness, for me, it seems to be counterintuitive. If we put all mass on a single action, it seems to be easier to learn than the case where we put even mass among actions.

3. In line 63, a near-optimal strategy, the wording strategy means an algorithm or a distribution? Based on footnote 1, it seems it means a distribution? so why in bandit, it needs to find a near-optimal strategy, it means the found distribution is very close to the true reward distribution in the underlying bandit problem?

4. Can you add some sketch about how to boost the constant probability ⅔ to $1-\delta$? It seems to be non-trivial and I guess we need to pay an extra log factor right?

5. For the bandit part, I feel that it is very close to the lower bound proof in bandit, but I am not sure how to connect them. So, can the authors explain more about the connection between your proposed approach and the exiting bandit lower bound?

6. I am not in this field but why do we need communication between prover and verifier?  And why RL also needs it? My understanding is, after the query, verifier knows the truth. So, they do not need to communicate at all. Why verifier need to tell something to the prover?

7. For the $\epsilon$-Nash equilibrium part, I feel that it is very close to differential privacy. Is it connected? My intuition is they both need to bound the probability rations.

8. For the corrupted bandit literature, I hope to see more discussions about the similarities and differences between corrupted bandit setting and the studied setting in this paper.

9. Typos: in Line 63 and 126, it should be an MAB not a MAB.

**Ethical Concerns:**

["NO or VERY MINOR ethics concerns only"]

**Final Justification:**

This work is well written. I do not have any further comments.

**Quality:**

4

**Strengths And Weaknesses:**

I am not an expert in the field of game theory; I provide my evaluation based on my own expertise in bandits.

Strengths: the studied problem is very interesting; it is not a typical problem set-up in bandit literature.

Weakness: some terminology are so confusing. If my understanding is right, for example, strategy and  policy here in this paper mean probability distribution, but in bandit/RL literature, it usually learning algorithms. Also, when I see VC, VC dimension hit my head.

For the presentation, I can follow most of the writings, but I suggest to move the formal problem set-up earlier.

Box below has my detailed questions.

---

> ### Author Rebuttal · Authors · 2025-07-29
>
> Thank you for carefully reading and evaluating our paper and for providing constructive feedback.
>
> Weaknesses
> ==
>
> > Some terminology are so confusing. If my understanding is right, for example, strategy and policy here in this paper mean probability distribution, but in bandit/RL literature, it usually learning algorithms. Also, when I see VC, VC dimension hit my head.
>
> Yes, we can totally understand the confusion! In game theory, a strategy or policy is usually a distribution over actions (for instance, people use the terms “pure strategy” and “mixed strategy”). As you point out, in bandits and RL the terminology is different. Because our paper covers both bandits and games, we decided to choose one terminology (the game theory one) and employ it throughout the paper.
>
> To reduce confusion, we will replace all appearances of “VC” in the text with the full phrase “vector commitment”.
>
> Questions
> ==
>
> > 1. Here, since some parts of this work use bandit. My first question is, from which aspects, the proposed problem set-up needs to balance the classical exploitation-vs-exploration tradeoff. I do not see any discussion about this trade-off through the paper. If it does not need to balance these two aspects, why using bandit?
>
> It is true that algorithms for bandits usually require balancing exploration vs. exploitation. In our paper, we are essentially approaching this problem from a slightly different perspective, asking, “what happens if we delegate the exploration to an untrusted party?” The prover claims they did all the exploration for us, and they are giving us a distribution of arms to pull that allegedly achieves (approximately) the best exploitation possible. Seeing as the prover is untrusted, can this information actually be useful? How many arm pulls does the prover save us compared to doing all the exploration ourselves?
>
> > 2. For the definition of sigma-smoothness, for me, it seems to be counterintuitive. If we put all mass on a single action, it seems to be easier to learn than the case where we put even mass among actions.
>
> You are correct that it is easier to find a strategy with high utility vs. finding a strategy with equally high utility that also satisfies an additional requirement of being smooth. Still, smoothness can be very desirable in some cases, e.g., because it makes the strategy more robust (“not putting all our eggs in one basket”), or because each arm can only be pulled a certain amount of times (each Uber driver can only do a certain number of rides a day), etc. An interesting point in our paper is that it is a lot easier to _verify_ optimality for smooth policies vs. general (non-smooth) policies.
>
>
> > 3. In line 63, a near-optimal strategy, the wording strategy means an algorithm or a distribution? Based on footnote 1, it seems it means a distribution? so why in bandit, it needs to find a near-optimal strategy, it means the found distribution is very close to the true reward distribution in the underlying bandit problem?
>
> In this paper, a policy and a strategy always mean the same thing: a probability distribution over the possible actions. This is consistent throughout the paper. See Definition 5.1. One way to find a near-optimal smooth strategy is to find a vector that is very close to the true utilities vector of the bandit, and then use Algorithm 2 (l. 836 in the appendix) to compute a strategy.
>
> > 4. Can you add some sketch about how to boost the constant probability ⅔ to $1-\delta$? It seems to be non-trivial and I guess we need to pay an extra log factor right?
>
> Definitely, and you are correct about the log factor. Given a protocol (V,P) with soundness and completeness ⅔ as in Def. 5.4, we can amplify this to $1-\delta$ as follows. Execute the protocol $k=O(\log(1/\delta))$ times. If V rejects more than $k/2$ times, then reject. Otherwise, for each (of the at least $k/2$) policies output by V, run the policy for $t=O(\log(k)/\epsilon^2)$ test steps. Finally, output the policy that obtained the highest utility during the test steps. Overall, if V makes $m$ oracle calls, then the amplified verifier uses $O(\log(1/\delta)*m + \log\log(1/\delta)/\epsilon^2)$ oracle calls.
>
> > 5. For the bandit part, I feel that it is very close to the lower bound proof in bandit, but I am not sure how to connect them. So, can the authors explain more about the connection between your proposed approach and the exiting bandit lower bound?
>
> Do you mean that the lower bound in Claim 2 (l. 644 in the appendix) is similar to the lower bound for learning bandits in Lemma 3 of Even-Dar et al. (2002)? That is true, as we point out in l. 864 of the appendix. In both cases, there is a reduction from the coin problem (Claim 10). The difference is that  Even-Dar et al. show a reduction from the coin problem to an algorithm that finds a good bandit arm. In our cases, we show a reduction from the coin problem to an interactive proof system with a prover and verifier that finds a _smooth_ bandit strategy.
>
> > 6. I am not in this field but why do we need communication between prover and verifier? And why RL also needs it? My understanding is, after the query, verifier knows the truth. So, they do not need to communicate at all. Why verifier need to tell something to the prover?
>
> You are correct that communication from the verifier to the prover is not strictly necessary, and indeed, in Thm 5.5 we present a protocol that consists of a single message from the prover to the verifier (the verifier does not send any messages). However, an interesting observation that we explore in this paper, is that the total number of bits of communication exchanged between the two parties can be significantly reduced if we allow the verifier to send messages to the prover as well.
> You can think of these messages sent by the verifier as *questions* that the prover must answer. The fact that the *verifier* chooses the questions is very useful, as if the prover chose the questions itself, it could always choose questions it can answer correctly.
>
> Our protocol for Thm 5.6 uses a number of communication bits that is sublinear in the number of actions n. The basic idea is to use a cryptographic commitment scheme: the prover sends a short commitment to the full proof, the verifier chooses some specific parts of the proof to inspect, and then the prover sends only those parts, and finally the verifier can check that what was sent matches the original commitment.
>
> > 7. For the Nash equilibrium part, I feel that it is very close to differential privacy. Is it connected? My intuition is they both need to bound the probability rations.
>
> Unfortunately we don’t quite see a connection to DP here. Perhaps if you elaborate further, things will click for us.
>
> > 8. For the corrupted bandit literature, I hope to see more discussions about the similarities and differences between corrupted bandit setting and the studied setting in this paper.
>
> Excellent question! In the corrupted bandit setting, a malicious adversary can change the information the learner gets when pulling an arm and thereby influence the learner’s estimate of the expectation of the arms.
> In our setting, we introduce a prover that may be adversarially controlled; however, the adversary has no control over the information that the verifier receives from the bandit itself. The prover should be thought of as a purported expert giving (possibly faulty) recommendations about the bandit. The learner (verifier) has access to these untrusted recommendations, as well as untampered-with access to the bandit.
> Our setting is more generous to the verifier than in the corrupted bandit setting: the verifier can always ignore the prover, in which case it is in the standard (non-corrupted) bandit setting. But the verifier can do better by not ignoring the prover.
>
> > 9. Typos: in Line 63 and 126, it should be an MAB not a MAB.
>
> Thanks for the catch! We will fix.

---

### Official Review · Reviewer_qtLM · 2025-07-13

**Clarity:** 3
**Significance:** 3
**Originality:** 3
**Rating:** 5
**Confidence:** 2

**Summary:**

This paper studies the problem of efficiently verifying the approximate optimality of strategies in multi-arm bandit and normal-form games under a smoothness constraint on the possible strategies. The paper considers $\sigma$-smooth strategies which are a set of randomized strategies where no action gets selected with probability more than $\sigma$.

In verification for MABs, instead of finding the optimal strategy, they consider the question that if a verifier is given a list of approximate rewards, can the verifier output a near optimal strategy or disprove that the given rewards are not approximately correct. The authors use the framework of interactive proof systems where a verifier interacts with an untrusted prover. They consider an oracle setting where the verifier and prover can both make queries to an oracle, and the goal is to find the approximate optimal strategy or reject the prover’s strategy by making sublinear in $n$ calls to the oracle.  $n$ is the number of possible arms in MAB, or the number of actions available to each player in normal-form games.

To achieve sublinear query complexities, protocols studied require $\sigma n = o(n)$ which is motivated as a requirement that the strategies considered are “well spread” on the support size.

The authors construct a protocol where $\varepsilon$-optimal $\sigma$ smooth bandit strategies can be verified where the verifier makes $\tilde O( \frac {n \sigma}{\varepsilon^2})$ calls to the oracle. If $\sigma n = o(n)$, then this is sublinear in $n$, which is lower the queries required to find the approximately optimal arm, i.e. $ \Omega (n)$ queries. The authors also show a matching  $ \Omega ( \frac {n \sigma}{\varepsilon^2})$ lower bound for the number of queries required under the set of protocols considered. The authors then use SNARK and vector commitment (VC) techniques from cryptography to lower the communication required using cryptography so that prover only has to communicate only sublinear in $n$ bits to the prover for the prover to be able to the find the approximately optimal smooth strategy.

For normal-form games, they use similar idea to to construct a proof for showing that a given strategy profile is an approximate smooth NE or reject if it's not approximately smooth, but now the verifier requires  $\tilde O( \frac {n k \sigma}{\eta^2})$ where $\eta$ is a slackness parameter.

**Questions:**

- Could you provide more intuition on when strategies are likely to be "smooth" in real-world scenarios other than the limited resource constraint example?
- Are there any broader class of problems in where the smoothness holds or is violated that can guide practitioners on when to use these protocols?
- Can you clarify the role of the slackness parameters and some intuitive discussion on why it’s needed for the game setting? Can you highlight the fundamental difference or challenge in adopting the results for the MAB case? Is the problem harder or is this a consequence of techniques?
- Can we verify the protocol empirically at least in the MAB setting?

**Ethical Concerns:**

["NO or VERY MINOR ethics concerns only"]

**Final Justification:**

The authors have responded to my concerns. I am increasing the score to an accept.

**Limitations:**

Yes

**Paper Formatting Concerns:**

No concerns.

**Quality:**

3

**Strengths And Weaknesses:**

Strengths:
- The paper introduces a novel problem, verifying strategy optimality over finding the optimal strategies in MAB and normal-form games.
- The problem studied is well motivated, studying smooth strategies has been motivated by previous works as well.
- The paper provides good theoretical foundations for the protocol provided. Under their setting, the protocol is able to accept or reject by making queries which are sublinear in $n$ and the authors provide a matching lower bound.
- Results for improving communication efficiency in the interactive proof and obtaining the approximately optimal strategy in sublinear in $n$ communication.
- Applications of these results for verifying approximate Nash equilibriums in games.

Weakness:
- Even though the requirements for  $\sigma n = o(n)$ are motivated through some examples, a more-in depth discussion of its implications is missing. A discussion of where it might hold or not hold could strengthen the general applicability of these protocols.

- The role of the slackness parameter is not well discussed in the paper. How should the slackness parameter be set with respect to $\varepsilon$?

- No empirical evidence for the claims.

---

> ### Author Rebuttal · Authors · 2025-07-29
>
> Thank you for carefully reading and evaluating our paper and for providing constructive feedback.
>
> On Smoothness
> ==
>
> > Even though the requirements for $\sigma n=o(n)$ are motivated through some examples, a more-in depth discussion of its implications is missing. A discussion of where it might hold or not hold could strengthen the general applicability of these protocols.
>
> > Are there any broader class of problems in where the smoothness holds or is violated that can guide practitioners on when to use these protocols?
>
> > Could you provide more intuition on when strategies are likely to be "smooth" in real-world scenarios other than the limited resource constraint example?
>
> Yes, absolutely!
>
> As you pointed out, one very general and common pattern that pops up in many real-world situations involves resource units with **bounded capacities** (what you called "limited resources"). These are scenarios where there is a system with many resource units, each of which can bear a limited load. For instance, one might want to assign jobs to computers in a data center, assign car traffic to various routes from point A to point B, assign containers to container ships, assign phone calls to agents in a call center, assign wireless communications to wave frequencies, etc. In each case, we want to assign each task or object to the best resource unit, but because each resource unit has only a limited capacity, the assignment will have to be smooth. In this sense, our setting is very broadly applicable.
>
> Going beyond the bounded capacities scenario, another very important consideration is **robustness**. Even if one could assign all tasks to a single resource unit, that is typically not advisable — it is not a good idea to “put all our eggs in one basket”. Even if the entire monthly batch of iPhones to be delivered to the USA from the factory in China could in principle fit in one container ship, it would still be a much better idea to distribute the load between a number of different ships, in order to limit the damage in case something goes awry. In that sense, requiring that a strategy be smooth is a way to ensure robustness.
>
> Smoothness is also desirable in **adversarial settings**, because it makes the strategy harder to anticipate. For instance, a security company might want to assign its patrols to the properties where burglaries are most likely (or more accurately —  properties where the expected loss is highest). But if it concentrates all its patrols on just a few properties, then burglars can simply focus their crimes on other properties. Therefore, it might make sense for the security company to use a smooth strategy that assigns patrols at random to many different properties under its responsibility, while still prioritizing properties according to expected loss.
>
> Lastly, smoothness also captures a certain sense of fairness or **equality**. In the foregoing example of a security company, if the company assigns all its patrols to a single house with the highest expected loss from burglary, that would be unfair towards the other home owners that pay for protection. On the other hand, if the company selects a perfectly smooth strategy, then it patrols all homes equally. In general, smoothness quantifies where the strategy falls on the spectrum between being fully equal and being fully unequal, with smoother strategies being more equal.
>
> On Empirical Evaluation
> ==
>
> > No empirical evidence for the claims.
>
> While implementation and evaluation are always important, we think it is reasonable for a theory paper focused on theoretical development like ours to not include an empirical section at least in the conference version. Please note that our paper is long (Full version is more than 40 pages) and is packed with new definitions that are already challenging to include within the 9 pages limit. There is no commonly agreed upon dataset such as imagenet or MNIST that allows for a simple evaluation of our protocol, and such an evaluation would likely need to consider both bandits and games making the paper significantly longer. Other influential theoretical papers in this area such as “the sample complexity of exploration in the multi-armed bandit problem” and “PAC Bounds for Multi-armed Bandit and Markov Decision Processes” do not include empirical evaluations as well. We think it is reasonable to provide such results for a journal version that is less space constrained. To be clear, we think implementing our protocols is certainly doable and cryptographic primitives used such as vector commitments and SNARKS have existing implementations that can be used towards this end, though this might be a big undertaking in practice.
>
> > Can we verify the protocol empirically at least in the MAB setting?
>
> Yes, this is definitely doable, but it is out of scope for a theoretical paper like ours.
>
> On Adopting to MAB
> ==
>
> > Can you highlight the fundamental difference or challenge in adopting the results for the MAB case? Is the problem harder or is this a consequence of techniques?
>
> We're not sure we understand this question. Could you please explain further? Our results are already for MAB (and games). Generally, we are not adopting existing results from some other setting to an MAB setting, so we don’t understand what you mean by “adopting the results for the MAB case”.

---

> > ### Comment · Reviewer_qtLM · 2025-08-06
> >
> > Thanks for answering my concerns.
> >
> > [1] On smoothness, I thank the authors for providing more examples where such strategies might be more desirable. As the authors also agree, the optimal smooth strategy may be far from the optimal. I wanted to get an understanding under which settings, the additional constraints of restricting to smooth strategies affects the optimal utility achievable. Is there a characterization of the optimal utility under the smoothness constraint vs when no such constraint is imposed?
> >
> > [2] On empirical evidence, I understand that this is mainly a theoretical contribution and is considered out of scope, but I still believe some empirical evidence could have highlighted the practicality of the approach.
> >
> > [3] on adopting to MAB:
> > My understanding is the MAB results are generalized/refined to obtain the results for the games. What are the main challenges in the proving the results for games and what aspect of the games setting makes it non trivial to "adopt" the MAB results to obtain the results for games?
> >
> > I will consider increasing my score in the discussion period.

---

> ### Author Response · Authors · 2025-08-08
> **Response**
>
> Thank you for these questions! Please see our answers below.
>
> > Is there a characterization of the optimal utility under the smoothness constraint vs when no such constraint is imposed?
>
> **Response:** Great question! As noted by the other reviewer there are examples where the optimal smooth policy is worse by an arbitrary large multiplicative factor. We will add such an example to the paper.
>
> Furthermore — yes! There is also a very simple characterization: Assume for simplicity that $\sigma = 1/k$ for some natural number $k$. Then it is easy to see that
>
> **Theorem**: the price of smoothness is $u_1 - \frac{1}{k}\sum_{i=1}^k u_i$, where $u_1 \geq u_2 \geq u_3 \dots$ are the utilities of the arms sorted in decreasing order.
>
> This is true simply because the optimal $\sigma$-smooth policy assigns equal weight of $1/k$ to the top $k$ arms, and therefore achieves utility which is the average of the top $k$ arms. We are very happy to add this characterization to the paper.
>
>
> > On empirical evidence, I understand that this is mainly a theoretical contribution and is considered out of scope, but I still believe some empirical evidence could have highlighted the practicality of the approach.”
>
> **Response:** We agree. However, as the paper is very loaded with new definitions and concepts, and as there are no existing benchmarks or datasets for this framework, we think it is unfeasible to add empirical evaluation to the current paper (also taking into account the 9-page limit). The best solution in our mind is to state empirical evaluation explicitly as a direction for future research.
>
> > on adopting to MAB: My understanding is the MAB results are generalized/refined to obtain the results for the games. What are the main challenges in the proving the results for games and what aspect of the games setting makes it non trivial to "adopt" the MAB results to obtain the results for games?”
>
> **Response:**
> The main challenge is that in the game setting, the prover has less choice over the strategy whose optimality they prove. This necessitates our introduction of a slackness parameter and our bandit protocol variant. We elaborate on this below.
> A second challenge is proving a stronger lower bound on the number of arm pulls of order $\Omega(k n \sigma)$, which requires new ideas that are not present in the bandit setting.
>
>
> **Slackness.** In the bandit setting, the prover can *choose* which policy to send to the verifier, as long as it is smooth and $\varepsilon$-close to optimal. This allows the prover to choose a policy $\pi$ whose value is very close (say, within $\varepsilon/4$) to optimal, creating slackness for the verifier’s approximation of its value. That is, as long as the verifier estimates the value of $\pi$ within additive error $3\varepsilon/4$, it can still determine it is $\varepsilon$-optimal.
> On the other hand, in the game setting, the prover is *given* a strategy profile, and it must prove that *this* strategy profile is an $\varepsilon$-approximate equilibrium. We do so by reducing this task to many bandit protocols. However, for these induced bandits the resulting policy may be *exactly* $\varepsilon$-optimal. In this case, there is no slackness for the verifier: It must exactly learn this policy’s value in order to ensure it isn’t $(\varepsilon + \delta)$-far from optimal for some arbitrarily small delta. This is impossible with a finite number of queries to the game oracle. To resolve this, we introduce the slackness parameter and construct a bandit protocol variant that takes as input a fixed policy, rather than allowing the prover to choose its own policy.

---

### Decision · Program_Chairs · 2025-09-17

**Decision:**

Accept (poster)

**Comment:**

This paper develops interactive-proof protocols for verifying the approximate optimality of σ-smooth strategies in multi-armed bandits and normal-form games using sublinear oracle queries. The key message in the paper is that verification can be more efficient than learning under smooth policies in both single and multi-agent settings. Moreover, the use of crypto-related techniques to reduce communication cost is also interesting.

Even though some reviewer has some negative comments (mainly on the presentation), after reading this paper myself, I think the criticisms are a little bit harsh.  To me, I find the newly introduced problem setting and framework very interesting, which might inspire many follow-up works in this direction. Thus, I recommend the acceptance. Meanwhile, I would strongly recommend that the authors take time to reorganize the structure and improve the presentation in the next version.